# Dataset Color Quantization: A Training-Oriented Framework for Dataset-Level Compression

**Chenyue Yu[3], Lingao Xiao[1,2,3], Jinhong Deng[1,2,4], Ivor W. Tsang[1,2,5], Yang He[1,2,3]**

[1]CFAR, Agency for Science, Technology and Research, Singapore
[2]IHPC, Agency for Science, Technology and Research, Singapore
[3]National University of Singapore
[4]University of Electronic Science and Technology of China (UESTC)
[5]Nanyang Technological University (NTU), Singapore

{e1143627, xiao_lingao}@u.nus.edu, jhdengvision@gmail.com
{Ivor.Tsang, He_Yang}@a-star.edu.sg

## Abstract

Large-scale image datasets are fundamental to deep learning, but their high storage demands pose challenges for deployment in resource-constrained environments. While existing approaches reduce dataset size by discarding samples, they often ignore the significant redundancy within each image – particularly in the color space. To address this, we propose Dataset Color Quantization (DCQ), a unified framework that compresses visual datasets by reducing color-space redundancy while preserving information crucial for model training. DCQ achieves this by enforcing consistent palette representations across similar images, selectively retaining semantically important colors guided by model perception, and maintaining structural details necessary for effective feature learning. Extensive experiments across CIFAR-10, CIFAR-100, Tiny-ImageNet, and ImageNet-1K show that DCQ significantly improves training performance under aggressive compression, offering a scalable and robust solution for dataset-level storage reduction.

## 1 Introduction

The rapid growth of datasets has been pivotal to the success of deep neural networks (DNNs) across diverse applications (Deng et al., 2009). However, storing and training on such datasets requires significant storage space and computing resources, which poses challenges for deployment on resource-constrained devices such as edge servers, drones, and industrial platforms (Mao et al., 2017). To alleviate this burden, dataset compression methods such as dataset pruning (Yang et al., 2022) and dataset distillation (Wang et al., 2018) have been widely studied. These approaches effectively reduce the number of training samples while maintaining recognition performance, yet they do not explicitly address the **per-sample storage overhead** that arises from representing full-color images.

In practice, image storage and transmission costs are often dominated by **color-space redundancy**. For example, images collected from remote monitoring or drone surveillance typically undergo color quantization to reduce bandwidth consumption. While color quantization (CQ; Heckbert (1982)) has been studied extensively for image compression and visualization (Puzicha et al., 2000), existing methods fall short in dataset-level training scenarios. Specifically, CQ can be categorized into two types: 1) **Image-Property-based CQ**: Preserving human visual perception according to the intrinsic properties of images by mapping the original color space to a reduced set of representative colors (Ozturk et al., 2014); and 2) **Model-Perception-based CQ**: Retaining recognizable features to enhance recognition accuracy by pre-trained neural networks (Hou et al., 2024).

Image-property-based CQ lacks neural network guidance, leading to two issues: (1) ambiguous semantic boundaries and insufficient color contrast (Kimchi & Peterson, 2008) that hinder discrim-

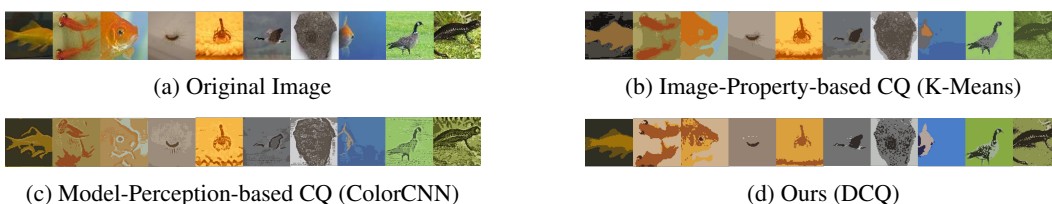

Figure 1: Visualization of different Color Quantization algorithms on Tiny-ImageNet. Images are quantized into 4 colors, which are 2 color bits. (a) Original images. (b) Color quantization is performed through K-Means clustering to obtain representative color palettes for each image, wasting bits on backgrounds. (c) Independent and representative color palettes obtained by ColorCNN, which have abrupt textural discontinuities. (d) Our DCQ assigns more colors to foregrounds and has less textural discontinuity.

inative feature learning; (2) uniform bit allocation across all colors, which wastes capacity on background while underrepresenting critical foreground features (Figure 1b).

In contrast, Model-perception-based CQ employs proxy networks to maintain recognition accuracy, but often introduces abrupt texture and edge discontinuities that distort visual features and degrade training performance (Geirhos et al., 2018). For instance, ColorCNN (Hou et al., 2020) quantizes CIFAR-10 images into 4 colors (2 bits) and achieves 77% accuracy under a pre-trained ResNet-18, yet training on the quantized dataset yields only 58% accuracy due to distorted textures (Figure 1c).

To address these limitations, we propose a unified Dataset Color Quantization (DCQ) framework that compresses datasets by reducing redundant color information while preserving both semantic and structural fidelity. Unlike existing methods that quantize images independently or optimize only for inference performance, our approach jointly considers dataset-level consistency, model-aware color significance, and visual structure preservation. Specifically, DCQ clusters images with similar color distributions to enable shared palette learning, prioritizes important regions based on model perception to retain critical semantics, and maintains texture continuity to avoid loss of structural information. This unified strategy achieves compact, quantization-aware datasets that remain effective for downstream training.

Our main contributions are summarized as follows: 1) To our knowledge, this is the first work to propose a solution using a limited set of color palettes to represent datasets, aiming to reduce storage requirements and enable training on color-restricted devices. 2) We introduce a dataset-level color quantization algorithm that combines cluster-shared color palettes, attention-guided bits allocation, and edge-preserving optimization. 3) Extensive experiments on various datasets, including CIFAR-10, CIFAR-100, Tiny-Imagenet, and ImageNet-1K, have validated the effectiveness of our method.

## 2 RELATED WORKS

### 2.1 COLOR QUANTIZATION

Color quantization reduces the number of colors used while preserving visual fidelity in RGB space. For maintaining human visualization, cluster algorithms such as MedianCut (Heckbert, 1982), OC-Tree (Gervautz & Purgathofer, 1988), and K-Means (Cheng & Wei, 2019) are common cluster algorithms used to group similar colors and represent each group with a single color to find the most representative color palettes; For maintaining neural network's recognition accuracy, ColorCNN (Hou et al., 2020) utilizes autoencoder to identify significant colors through end-to-end training processing for each image. ColorCNN+ (Hou et al., 2024) enhances traditional color quantization by combining deep learning and clustering techniques. CQFormer (Su et al., 2023) performs color quantization by mapping input images to quantized color indices through its Annotation Branch, while its Palette Branch identifies key colors in the RGB space. Although these algorithms work well with neural networks and preserve human visual recognition, they do not effectively optimize the train-set for better model performance.

## 2.2 DATASET PRUNING

Dataset Pruning, also known as Coreset Selection, aims to shrink the storage of a dataset by selecting important samples according to some predefined criteria. GraNd/EL2N (Paul et al., 2021) quantizes the importance of a sample with its gradient magnitude. TDDS (Zhang et al., 2024) uses a dual-depth strategy to achieve a balance between incorporating extensive training dynamics and identifying representative samples for dataset pruning. CCS (Zheng et al., 2022) proposed a stratified sampling strategy to increase the classification accuracy in a high pruning ratio. Entropy (Coleman et al., 2019) proposes selecting the highest-entropy examples for the coreset, as entropy captures the uncertainty of training examples. Forgetting (Toneva et al., 2018) refers to the number of instances where an example is misclassified after it has previously been correctly classified during model training and AUM (Pleiss et al., 2020) is a data difficulty metric that identifies mislabeled data. However, these algorithms experience a significant accuracy drop at high compression rates, necessitating alternative methods for effective dataset compression under such conditions which means we need to pioneer a new direction to solve this problem, not just reduce the storage by removing images.

## 3 METHODOLOGY

### 3.1 MOTIVATION

Dataset pruning and distillation reduce sample counts to lower storage and computation costs, but often overlook per-sample redundancy. In images, storage and transmission are dominated by color redundancy: many pixels share nearly identical colors, especially in smooth regions (e.g., skies, walls) or gradual textures. For instance, a 256×256 RGB image has over 65k pixels but only a few thousand distinct colors. Thus, much capacity is wasted on subtle, semantically irrelevant variations, motivating color quantization or palette-based compression to cut per-image storage while preserving semantic content.

To reduce the color-space redundancy, Color quantization (CQ) has been extensively employed in the contexts of image compression and visualization. However, existing CQ techniques are ill-suited for dataset-level training. Formally, we define color quantization as follows:

**Definition 1** (Color Space Distribution). *Let $\mathcal{S} \subseteq \mathbb{R}^3$ be the RGB color space. We define two fundamental color distribution representations:*

*(i) Original Color Palette $\mathcal{P} = \{p_i \in \mathcal{S}\}_{i=1}^m$, where each $p_i$ represents a unique RGB color vector in the continuous color space. (ii) Quantized Color Palette $\mathcal{C} = \{c_i \in \mathbb{R}^3\}_{i=1}^k$, where $k \ll m$, and each $c_i$ represents a representative color vector that preserves the most discriminative color features from the original distribution $\mathcal{P}$. Here $d$ denotes the dimensionality of the target color space. The mapping from $\mathcal{P}$ to $\mathcal{C}$ defines a color quantization function:*

$$Q : \mathcal{P} \to \mathcal{C}$$

*where $Q$ performs both cardinality reduction and potential space transformation while maintaining essential color characteristics. Given an image $I$ with original palette $\mathcal{P}$, the quantized image $I_Q$ is obtained through:*

$$I_Q = Q(I; \mathcal{C}) : I \times \mathcal{C} \to I_Q$$

*where $I_Q$ represents the reconstructed image using only colors from palette $\mathcal{C}$.*

Existing CQ methods can be broadly categorized into two paradigms: 1) Image-Property-Based CQ: This approach reduces color-space redundancy by grouping chromatically similar pixels. While computationally straightforward, it is inherently limited in semantic preservation. Gradual color transitions spanning multiple objects or regions may cause semantically distinct areas to be erroneously merged, resulting in a loss of structural information and critical boundary details. 2) Model-Perception-Based CQ: These methods leverage pre-trained neural networks to guide color allocation (Hou et al., 2020), ensuring high recognition accuracy for quantized images. However, they often introduce abrupt texture and edge discontinuities, which can distort visual structure and impair the learning of complete feature representations (Geirhos et al., 2018).

Crucially, both paradigms emphasize **inference-oriented performance**, optimizing the recognition of pre-trained models. Formally, given a dataset $\mathcal{D} = \{(x_i, y_i)\}_{i=1}^N$, a quantization function $Q(\cdot)$,

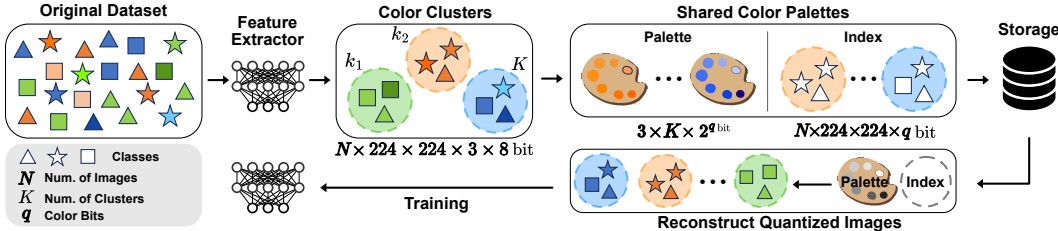

Figure 3: The pipeline of our dataset color quantization framework. First, we apply K-means clustering to group images based on their features extracted from a pre-trained model. Next, within each cluster, we perform K-means on the color palettes of individual images to generate a shared color palette for all images in the same cluster. The generated palettes and their corresponding indices are stored for later use. During training, we retrieve the stored indices and palettes to reconstruct quantized images, which are then used to train a neural network.

and a fixed pretrained model $f_\phi$, inference-oriented quantization optimizes:

$$Q^\star = \arg\max_Q \ \mathbb{E}_{(x,y)\sim\mathcal{D}}\big[\mathcal{A}(f_\phi(Q(x)), y)\big], \tag{1}$$

where $\mathcal{A}$ denotes the evaluation metric (*e.g.*, accuracy).

In contrast, our dataset-level color quantization is to optimize the training of high-performance models on quantized datasets. Formally, let $\mathcal{D}_{\text{train}}$ and $\mathcal{D}_{\text{test}}$ denote the training and test sets. Quantization induces a training set $\mathcal{D}_Q = \{(Q(x), y)\}$. A model $f_\psi$ trained on $\mathcal{D}_Q$ is obtained as:

$$\psi^\star(Q) = \arg\min_\psi \ \mathbb{E}_{(x,y)\sim\mathcal{D}_Q}\big[\ell(f_\psi(Q(x)), y)\big]. \tag{2}$$

The training-oriented quantizer is then:

$$Q^\star_{\text{train}} = \arg\max_Q \ \mathbb{E}_{(x,y)\sim\mathcal{D}_{\text{test}}}\big[\mathcal{A}(f_{\psi^\star(Q)}(x), y)\big]. \tag{3}$$

To overcome these limitations, we propose a novel **Dataset Color Quantization (DCQ)** framework, which integrates color quantization into dataset-level compression. Unlike conventional CQ methods that operate independently on individual images, DCQ jointly optimizes palette sharing and semantic preservation across the dataset, achieving a substantial reduction in storage while retaining training efficacy. Specifically, we first design a Chromaticity-Aware Clustering (CAC) strategy to aggregate images with chromatically similar distributions into clusters and construct a shared cluster-level palette instead of a sample-wise color palette in previous CQ methods. Then, we propose an Attention-Guided Palette Allocation mechanism that leverages model-derived attention to prioritize color representation for semantically critical regions, ensuring the preservation of essential object features.

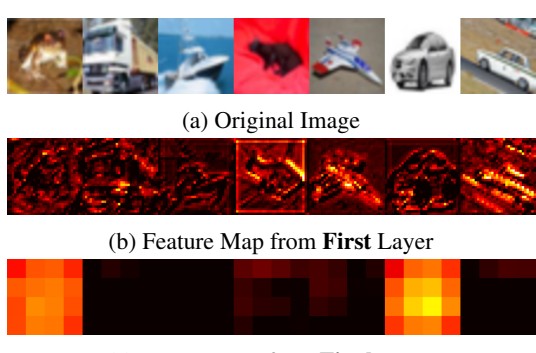

(a) Original Image

(b) Feature Map from **First** Layer

(c) Feature Map from **Final** Layer

Figure 2: Comparison of the original image and feature maps extracted from a ResNet-18 trained on CIFAR-10. Thermal color maps visualize activation strength from black to white, reflecting learned feature hierarchy.

Moreover, we leverage differentiable quantization to refine the color palette to retain edge and texture fidelity, preserving structural cues crucial for downstream training. Through these mechanisms, DCQ provides a principled approach to dataset-level color compression, effectively minimizing redundant information while safeguarding both semantic and structural content indispensable for high-fidelity model training.

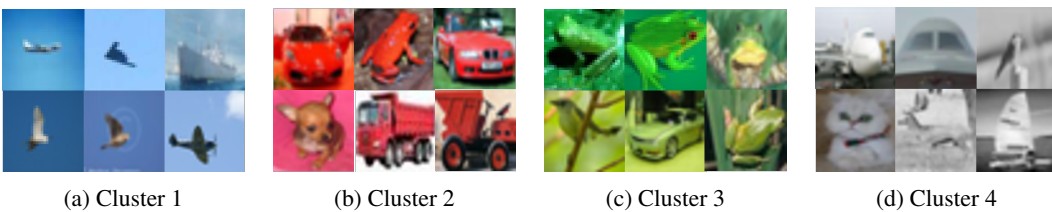

| (a) Cluster 1 | (b) Cluster 2 | (c) Cluster 3 | (d) Cluster 4 |

Figure 4: Visualization of four clusters obtained by applying ResNet-18 first-block features to partition CIFAR-10 into 20 clusters.

## 3.2 DATASET COLOR QUANTIZATION

The pipeline of the Dataset Color Quantization (DCQ) framework is shown in Figure 3. DCQ first groups images with chromatically similar distributions into clusters and constructs a shared cluster-level palette. Next, within each cluster, we perform K-means on the color palettes of individual images to generate a shared color palette for all images in the same cluster. The generated palettes and their corresponding indices are stored for later use. During training, we retrieve the stored indices and palettes to reconstruct quantized images, which are then used to train a neural network. In the following, we will give a detailed introduction of the proposed method.

### 3.2.1 CHROMATICITY-AWARE CLUSTERING

Traditional CQ methods typically construct palettes independently for each image. This per-image quantization strategy allows the color palette to be optimized to the specific distribution of a single image, thereby minimizing quantization error within that image. However, such independence introduces inconsistency across the dataset: visually similar regions in different images may be mapped to disparate representative colors. As a result, the color space perceived by the training model becomes fragmented, which increases the difficulty of learning stable semantic boundaries and weakens the model's generalization ability.

To overcome this limitation, we propose the use of shared palettes across multiple images. By quantizing a collection of images rather than on individual samples, the learned palettes achieve greater cross-image consistency. This consistency reduces the semantic ambiguity induced by palette misalignment and enables the model to perceive color distributions more stably. Nevertheless, constructing a single palette from a large set of diverse images poses a new challenge. When the overall color distribution becomes too broad, a fixed-size palette must accommodate a wider variety of colors. Consequently, the quantization error for individual images increases, as the palette can no longer precisely capture each image's local chromatic distribution.

To this end, we introduce Chromaticity-Aware Clustering (CAC) mechanism that first partitions the dataset into groups of images with similar color distributions to balance the trade-off between cross-image consistency and quantization fidelity. Shared palettes are then generated within each cluster, allowing the method to retain cross-image consistency while constraining the color variance that the palette must represent. In this manner, quantization error remains controlled, and the resulting palettes better preserve semantic boundaries and structural features across images.

In particular, to identify images with similar color distributions, we utilize semantic feature maps $\psi_i(x)$, which are the outputs of the $i$-th layer in a neural network model $\mathcal{M}$. These feature maps capture hierarchical representations of an image, from low-level color patterns to high-level semantics. The resulting feature vector provides a comprehensive representation of the image's color distribution, enabling effective comparison across images. While deeper output feature maps $\psi_i(x)$ encode abstract semantic concepts like object parts and contextual relationships, shallow layers capture local patterns including color distributions (Zeiler & Fergus, 2014).

Through empirical analysis of feature distributions across layers, we observe an inherent trade-off:

$$\text{as } i \uparrow: \begin{cases} \text{Sem}(\psi_i) \uparrow & \text{(semantic abstraction)} \\ \text{Vis}(\psi_i) \downarrow & \text{(visual fidelity),} \end{cases} \quad (4)$$

where Sem($\cdot$) represents semantic information capacity and Vis($\cdot$) indicates visual fidelity, and the visualization of feature-map is shown in Figure 2. Based on this observation, we select the shallow layer feature map as our perceptual representation:

$$\phi_{\mathcal{M}}(x) = \psi_{shallow}(x) \tag{5}$$

We partition the dataset using K-Means clustering on semantic features $\psi_{shallow}(x)$ to create $k$ clusters. We set $k = 20$ through ablation experiments. As shown in Figure 4, this clustering not only groups images by dominant color distributions (blue, red, green, and gray tones) but also transcends class boundaries by grouping semantically distinct objects with similar chromatic properties.

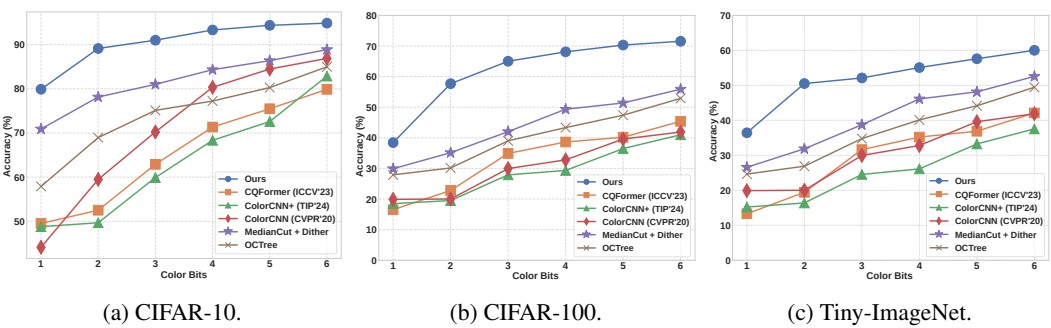

(a) CIFAR-10.  (b) CIFAR-100.  (c) Tiny-ImageNet.

Figure 5: Comparison between prior color quantization methods and our approach, evaluated with ResNet-18. Detailed results are provided in Appendix B. Unlike the original paper, which quantized the test set while training on the original train set, we quantize the entire train-set and keep the test-set unchanged.

### 3.2.2 ATTENTION-GUIDED PALETTE ALLOCATION

Traditional color quantization approaches typically assume uniform importance across color palettes (Celebi, 2023). However, different image regions contribute unequally to semantic recognition, and allocating bits uniformly across all regions may neglect critical structures. To address this, we introduce the notion of *Color Palette Impact Level*, which distinguishes palettes that are crucial for model performance from those that are less influential.

**Definition 2** (Color Palette Impact Level). *We categorize the roles of color palettes in model recognition into two distinct classes: (1) **High-Impact Palettes** ($P_H$): These palettes significantly influence model performance. Changing them to any other palette results in a noticeable increase in loss. (2) **Low-Impact Palettes** ($P_L$): These palettes have little effect on model performance. There exists at least one alternative palette that can replace them without a meaningful change in loss.*

To identify high-impact palettes, we employ Grad-CAM++ (Chattopadhay et al., 2018) to obtain an attention map generated from task-specific pre-trained models. The Grad-CAM++ heatmap reveals discriminative regions that contribute to the network's classification decision. The higher attention values indicate greater relevance to the network's prediction. For each image, we retain the top $k_{Gra}\%$ of pixels with the highest attention values, ensuring that only the most relevant regions are preserved, while all remaining pixels are set to zero. The value of $k_{Gra}\%$ was determined through ablation studies in Appendix C.4.

The selected pixels from all images within a cluster are aggregated to form an expanded palette space. Following prior work (Orchard et al., 1991), we convert the RGB values of these pixels to LAB color space to better preserve perceptual similarity. Finally, K-means clustering is applied to this aggregated LAB palette space to generate a shared quantized palette that balances compactness and semantic fidelity across images.

### 3.2.3 TEXTURE-PRESERVED PALETTE OPTIMIZATION.

A key goal of color quantization is to preserve important texture information. However, traditional K-Means clustering assigns pixels based solely on color similarity, ignoring the image's structural

details. Without an error feedback mechanism to measure Texture Loss (TL), particularly in local gradients, edges, and spatial patterns, this approach often leads to texture degradation, especially in fine-detail regions. To address this, our algorithm refines K-Means-generated color palettes through a quantizable optimization process, reducing Texture Loss (TL) and better preserving local structural details.

Inspired by style transfer (Johnson et al., 2016), we designed a differentiable textural palette optimization model that performs differentiable color quantization by mapping each pixel in the image to its nearest color in a palette, applying a straight-through estimator (STE) (Yin et al., 2019) to enable backpropagation through the quantization process. For each cluster, after obtaining the shared color palette, we optimize the palette using a gradient descent algorithm to minimize the loss function, ensuring that the color differences between the original and quantized images are minimized. We optimize the color palette by minimizing the edge distribution differences $EL$ between the original image and image after color quantization which can be calculated using:

$$
\begin{aligned}
G(I) &= \sqrt{(I * S_x)^2 + (I * S_y)^2}, \\
EL &= \sum_{i=1}^{3} w_i \cdot \mathrm{MSE}\Big( G(I_{\mathrm{orig}}^i), G(I_{\mathrm{quant}}^i) \Big),
\end{aligned}
\tag{6}
$$

where $I_{\mathrm{orig}}^i$ and $I_{\mathrm{quant}}^i$ represent the $i$-th channel (L, A, B) of the original and quantized images, $S_x$ and $S_y$ using Sobel operator (Kanopoulos et al., 1988) to calculate edge information for each channel, MSE stands for the mean squared error and $w_i$ is the weights for each channel and $*$ stands for convolution, adjusting the contribution of each channel to the total loss. The result of using Texture-preserved Palette Optimization was shown as ablation experiments in Appendix C.1.

Figure 1 shows the result of our dataset color quantization algorithm. Under 2-bit color quantization, compared to traditional Image-Property-based quantization, our approach achieves superior edge-texture preservation and palette allocation efficiency, resulting in enhanced structural integrity and detail fidelity. Compared to Model-Perception-based quantization, our method better preserves perceptual fidelity with enhanced texture continuity.

## 4 EXPERIMENTS

### 4.1 EXPERIMENT SETTINGS

In this task, the effectiveness of the proposed dataset color quantization is evaluated from popular datasets: CIFAR-10, CIFAR-100 (Krizhevsky et al., 2009), Tiny-ImageNet (Le & Yang, 2015) and ImageNet-1K (Deng et al., 2009) with ResNet (He et al., 2016). Details of datasets are provided in the Appendix A.1. In our algorithm, we used our color quantization algorithm on the original train-set and kept original test-set. Since our work tackles a less-studied problem of dataset color quantization with no known clear solution, it is important to set an adequate baseline for comparison. We choose two aspects of the most relevant baselines to show the efficiency of our algorithms. (1) For **color quantization**, we choose ColorCNN (Hou et al., 2020), ColorCNN+ (Hou et al., 2024), CQFormer (Su et al., 2023), MedianCut (Heckbert, 1982) and OCTree (Gervautz & Purgathofer, 1988). We apply image color quantization algorithms to every image in the entire train-set while keeping the test-set unchanged. This differs from the original paper, which quantized the test-set while training on the original train-set. (2) For **dataset pruning** algorithms, EL2N (Paul et al., 2021), Entropy (Coleman et al., 2019) and Forgetting (Toneva et al., 2018), CCS (Zheng et al., 2022), TDDS (Zhang et al., 2024) are used as baselines. These algorithms use a coreset from the train-set to train a model and evaluate it on the original test-set. More information about baselines is provided in the Appendix A.2.2. The compression ratio achieved through color quantization can be formally defined in relation to the bit depth reduction. Starting from the standard 24-bit RGB color space (8 bits per channel), when quantizing to $q$ bits, the color palette is reduced to $2^q$ distinct colors. The corresponding compression ratio $q_r$ is given by $q_r = 1 - q/24$, establishing a direct mapping between the compression levels of color quantization and dataset pruning strategies.

**Comparison to Color Quantization.** Figure 5 shows the performance of using the color quantized dataset for training. Our performance substantially outperforms other methods from 1 to 6 bits for all

Table 1: Comparison of dataset pruning algorithms and our dataset color quantization algorithm on CIFAR-10, CIFAR-100 with ResNet-18, and ImageNet-1K with ResNet-34.

| | CIFAR-10 | | | | | CIFAR-100 | | | | | ImageNet-1K | | | | |
|---|---|---|---|---|---|---|---|---|---|---|---|---|---|---|---|
| Color Bits | 5 | 4 | 3 | 2 | 1 | 5 | 4 | 3 | 2 | 1 | 5 | 4 | 3 | 2 | 1 |
| Colors/Image | 32 | 16 | 8 | 4 | 2 | 32 | 16 | 8 | 4 | 2 | 32 | 16 | 8 | 4 | 2 |
| Prune Ratio | 80% | 83% | 87.5% | 92% | 96% | 80% | 83% | 87.5% | 92% | 96% | 80% | 83% | 87.5% | 92% | 96% |
| Random | 88.15 | 84.38 | 80.15 | 77.04 | 70.08 | 57.36 | 50.96 | 42.19 | 39.71 | 36.68 | 62.54 | 58.18 | 53.34 | 50.32 | 20.28 |
| Entropy | 79.08 | 74.59 | 67.58 | 64.51 | 57.06 | 41.83 | 35.45 | 29.77 | 28.96 | 22.16 | 55.80 | 44.59 | 42.04 | 31.04 | 17.06 |
| EL2N | 70.32 | 67.06 | 25.99 | 21.31 | 19.85 | 15.51 | 12.51 | 10.36 | 8.36 | 7.51 | 31.22 | 21.28 | 13.99 | 11.69 | 9.81 |
| AUM | 57.84 | 49.09 | 29.58 | 25.60 | 21.35 | 16.38 | 11.83 | 9.45 | 8.77 | 5.37 | 21.12 | 15.09 | 10.13 | 8.93 | 6.35 |
| CCS$_{AUM}$ | 90.53 | 88.97 | 87.61 | 75.01 | 73.02 | 63.19 | 60.05 | 52.16 | 30.02 | 28.24 | 64.49 | 58.44 | 57.96 | 45.58 | 31.31 |
| CCS$_{Forg.}$ | 90.93 | 88.05 | 86.66 | 74.31 | 73.02 | 63.99 | 60.45 | 51.86 | 31.12 | 27.33 | 65.01 | 58.74 | 57.57 | 46.28 | 30.44 |
| CCS$_{EL2N}$ | 89.81 | 88.47 | 87.01 | 74.77 | 73.17 | 61.83 | 60.75 | 52.16 | 32.07 | 28.17 | 64.71 | 58.15 | 56.54 | 45.19 | 29.14 |
| TDDS (Strategy-E) | 91.30 | 88.72 | 87.47 | 77.32 | 72.46 | 63.01 | 61.44 | 55.13 | 32.15 | 26.55 | 62.56 | 56.48 | 54.91 | 43.91 | 29.56 |
| **DCQ (Ours)** | **94.39** | **93.34** | **91.02** | **89.15** | **79.90** | **69.05** | **67.89** | **65.02** | **57.69** | **38.44** | **66.99** | **64.34** | **62.02** | **49.69** | **35.95** |
| Full Accuracy | 95.45% | | | | | 78.21% | | | | | 73.54% | | | | |

Table 2: Ablation Studies of Different Strategies.

(a) Clustering Features Comparison.

| Bits | Label | Random | Image | Final | Shallow |
|---|---|---|---|---|---|
| 1 | 40.10 | 28.44 | 68.44 | 42.10 | **79.90** |
| 2 | 51.08 | 39.65 | 79.15 | 53.78 | **89.15** |
| 3 | 65.33 | 53.21 | 81.21 | 66.39 | **91.02** |
| 4 | 74.05 | 61.33 | 84.33 | 75.15 | **93.34** |
| 5 | 79.42 | 70.29 | 89.29 | 80.44 | **94.39** |
| 6 | 81.99 | 75.52 | 90.52 | 83.93 | **94.89** |

(b) Cluster Numbers on CIFAR-10 (First Level clustering).

| Bits | 1 | 2 | 3 | 4 | 5 | 6 |
|---|---|---|---|---|---|---|
| 1 cluster | 71.02 | 84.52 | 86.71 | 89.49 | 90.55 | 91.01 |
| 10 cluster | 76.12 | 87.95 | 90.08 | 92.14 | 92.79 | 93.69 |
| **20 cluster** | **79.90** | **89.15** | **91.02** | **93.34** | **94.39** | **94.89** |
| 50 cluster | 77.34 | 87.23 | 90.11 | 92.53 | 92.41 | 93.05 |
| 100 cluster | 76.94 | 86.61 | 90.91 | 91.83 | 92.66 | 93.11 |
| 150 cluster | 76.77 | 86.21 | 87.99 | 90.04 | 92.31 | 93.58 |

(c) Attention Map Methods (Second level clustering).

| Bits | 1 | 2 | 3 | 4 |
|---|---|---|---|---|
| None | 72.02 | 84.02 | 86.79 | 89.09 |
| GradCAM | 75.05 | 85.75 | 88.02 | 90.55 |
| **GradCAM++** | **79.90** | **89.15** | **91.02** | **93.34** |
| RISE | 79.04 | 88.91 | **92.05** | 93.32 |
| LayerCAM | **80.66** | 89.71 | 90.95 | 93.09 |

three datasets including (a) CIFAR-10, (b) CIFAR-100, and (c) Tiny-ImageNet. Using a 2-bit color palette (4 colors) for quantization, our method (DCQ) achieves accuracies of 89.15% on CIFAR-10, 57.69% on CIFAR-100, and 50.51% on Tiny-ImageNet. This represents a improvement over the ColorCNN (Hou et al., 2020) results, with gains of 30.00% (89.15% - 59.15%) on CIFAR-10, 35.37% (57.69% - 22.32%) on CIFAR-100, and 18.06% (50.51% - 32.45%) on Tiny-ImageNet.

**Comparison to Dataset Pruning.** Table 1 presents the performance comparison of various dataset pruning methods against our dataset color quantization (DCQ) algorithm. DCQ consistently outperforms other methods across all pruning ratios and datasets. For instance, under a 96% compression ratio, we apply 1-bit quantization (2 colors per image), clustering the entire dataset into 20 clusters. As a result, only 40 distinct colors are used to quantize the entire train-set. DCQ achieves accuracies of 79.9% on CIFAR-10 and 38.44% on CIFAR-100, significantly surpassing the CCS results of 73.02% and 28.24%, respectively. These results highlight DCQ as an effective color quantization strategy, particularly under high pruning ratios. Similarly, DCQ excels on ImageNet-1K under high pruning ratios.

## 4.2 ABLATION STUDIES

**How Many Clusters?** Table 2b presents the impact of varying the number of clusters in the initial K-Means clustering step. Using the least number of clusters, which is 1 cluster, extracts all color palettes to get a shared quantized color palette for every image in the dataset. On the other hand, the most number of clusters (i.e., 50,000 clusters for CIFAR-10) means that we assign each image a unique quantized color palette, which can be seen as Image-Property-based CQ. The best performance is achieved with 20 clusters.

Table 3: Comparison of dataset pruning algorithms and our algorithm on CIFAR-10 with ResNet-34 and ResNet-50.

| | ResNet-34, CIFAR-10 | | | | | ResNet-50, CIFAR-10 | | | | |
|---|---|---|---|---|---|---|---|---|---|---|
| | 80% | 83% | 87.5% | 92% | 96% | 80% | 83% | 87.5% | 92% | 96% |
| Random | 86.13 | 83.48 | 76.19 | 74.44 | 65.08 | 87.03 | 83.18 | 79.19 | 75.44 | 68.09 |
| EL2N | 70.21 | 66.36 | 24.69 | 23.11 | 22.19 | 69.71 | 65.96 | 24.59 | 23.31 | 21.89 |
| AUM | 58.34 | 49.59 | 29.98 | 21.60 | 20.35 | 57.84 | 49.09 | 29.58 | 25.60 | 21.35 |
| CCS$_{AUM}$ | 89.34 | 87.92 | 87.52 | 74.01 | 71.42 | 89.44 | 88.11 | 86.52 | 74.31 | 71.02 |
| TDDS | 89.58 | 88.15 | 86.92 | 73.05 | 69.48 | 90.68 | 88.55 | 87.62 | 74.01 | 70.46 |
| **Ours** | **94.39** | **93.14** | **91.15** | **89.47** | **79.87** | **94.19** | **92.94** | **90.55** | **88.27** | **77.26** |

**Which Feature to Use?** Table 2a compares clustering performance across different feature extraction approaches: (1) label-based grouping that clusters images within the same class; (2) randomly

assign images into different clusters; (3) direct image clustering that applies K-Means on raw pixel values; (4) final-layer feature maps from the final residual block and (5) shallow-layer feature maps from the first residual block as the different baselines. The shallow-layer features consistently outperform all alternatives, which aligns with our analysis in Sec. 3.2.1. Table 2a shows that clustering with Shallow-layer Feature maps achieves the best results compared to other features.

**Which Method to Get Attention Maps?** Table 2c compares different attention map generation methods, including Grad-CAM (Selvaraju et al., 2016), LayerCAM (Jiang et al., 2021), and RISE (Petsiuk et al., 2018). Our algorithm is guided by attention maps and is not limited to Grad-CAM++; other methods are equally applicable.

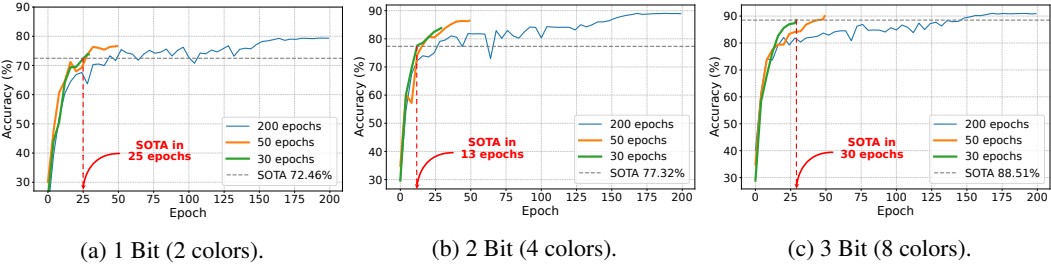

(a) 1 Bit (2 colors).      (b) 2 Bit (4 colors).      (c) 3 Bit (8 colors).

Figure 6: Illustration of the test accuracy of our method using ResNet-18 on CIFAR-10.

## 4.3 ADDITIONAL ANALYSIS

**Training Efficiency.** Figure 6 illustrates a key limitation of traditional dataset pruning. Although pruning reduces the dataset from $N$ to $\alpha N$ samples ($\alpha < 1$), the total number of parameter updates remains nearly unchanged to ensure convergence. With batch size $b$, the original dataset requires $N/b$ updates per epoch, while the pruned dataset requires $N/(\alpha b)$ to match optimization steps. This compensatory mechanism cancels potential computational gains, so reduced dataset size does not improve training efficiency. Hence, our comparison at the same compression ratio in Table 1 is fair.

Table 4: Combining color quantization and dataset pruning for extreme compression ratio. The pruning rate is 90%, and color quantization is $q$-bit.

Table 5: Comparison of dataset distillation algorithms and our algorithm on different datasets using ResNet-18. Entries with OOM denotes Out of Memory.

| Color Bits | CIFAR-10 | | | | CIFAR-100 | | | |
|---|---|---|---|---|---|---|---|---|
| | $q=5$ | $q=4$ | $q=3$ | $q=2$ | $q=5$ | $q=4$ | $q=3$ | $q=2$ |
| Total Ratio | 98.0% | 98.4% | 98.7% | 99.2% | 98.0% | 98.4% | 98.7% | 99.2% |
| Random | 55.13 | 50.48 | 46.19 | 31.44 | 27.03 | 23.34 | 19.18 | 15.47 |
| EL2N | 17.11 | 16.36 | 14.69 | 13.11 | 7.01 | 6.94 | 5.38 | 4.33 |
| CCS$_{AUM}$ | 64.20 | 59.69 | 49.35 | 46.25 | 23.14 | 22.58 | 21.53 | 20.11 |
| TDDS | 63.28 | 55.18 | 46.92 | 43.05 | 22.66 | 21.52 | 20.62 | 18.01 |
| **Ours + CCS$_{AUM}$** | **79.92** | **79.53** | **76.01** | **70.73** | **46.26** | **40.22** | **39.49** | **30.94** |

| | CIFAR-10 | | Tiny-ImageNet | | ImageNet-1K | |
|---|---|---|---|---|---|---|
| | IPC=50 | IPC=10 | IPC=100 | IPC=50 | IPC=100 | IPC=50 |
| DM | 63.0 | 53.9 | OOM | OOM | OOM | OOM |
| MTT | 71.6 | 65.3 | 28.0 | OOM | OOM | OOM |
| SRe2L | 47.5 | 27.2 | 49.7 | 41.1 | 52.8 | 46.8 |
| G-VBSM | 59.2 | 53.5 | 51.0 | 47.6 | 55.7 | 51.8 |
| **Ours** | **70.9** | **67.4** | **57.6** | **50.5** | **59.6** | **54.4** |

**Network Generalization.** Table 3 shows that our algorithm outperforms others on larger networks (ResNet-34, ResNet-50), demonstrating strong transferability across architectures. Results for additional networks are provided in Appendix C.2.

**Combine with Dataset Pruning and Comparison with Dataset Distillation.** Since our method is orthogonal to dataset pruning, it can be combined with pruning to achieve higher compression. Table 4 reports results of integrating CCS-based coreset selection (10% retention; Zheng et al. (2022)) with $q$-bit color quantization, reaching extreme ratios up to 99.2% (70.73% accuracy on CIFAR-10), which highlight the effectiveness of hybrid strategies for extreme data compression. For dataset distillation, we use G-VBSM (Shao et al., 2024), SRe2L (Shao et al., 2024), MTT (Cazenavette et al., 2022), and DM (Cazenavette et al., 2022) as baselines. To match the compression rates in Shao et al. (2024), we apply CCS$_{AUM}$ with a pruning ratio $r\%$ and quantize the coreset to $q$ bits. For CIFAR-10 (IPC = 10, 99.8% compression; IPC = 50, 99%), we set $(r\%, q)$ to (95%, 1) and (75%, 1). For Tiny-ImageNet (IPC = 50, 90%; IPC = 100, 80%), we set $(r\%, q)$ to (0%, 5) and (0%, 2). For ImageNet-1K (IPC = 50, 96.1%; IPC = 100, 92.1%), we set $(r\%, q)$ to (50%, 2) and (60%, 5). This maintains consistency with the original paper. Table 5 shows our algorithm outperforms others.

**Additional Analysis and Visualization.** Additional analysis is provided in Appendix D, and visualizations of all datasets are provided in the Appendix E.

## 5 CONCLUSION

We present a novel dataset condensation approach that leverages color quantization to reduce storage, improving data efficiency and enabling new forms of compact data representation. While effective, our work has several limitations. Future research could investigate adaptive, per-image quantization strategies to balance compression and accuracy more flexibly. In addition, developing neural architectures specifically optimized for color-quantized data, rather than full-color inputs, may further enhance performance on compressed datasets. These directions highlight the potential of integrating quantization with dataset-level learning to advance efficient and robust deep learning.

## ACKNOWLEDGEMENT

This research is supported by A*STAR Career Development Fund (CDF) under Grant C243512011, the National Research Foundation, Singapore under its National Large Language Models Funding Initiative (AISG Award No: AISG-NMLP-2024-003). Any opinions, findings and conclusions or recommendations expressed in this material are those of the author(s) and do not reflect the views of National Research Foundation, Singapore.

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

# A EXPERIMENTS SETUP

## A.1 DATASETS

**CIFAR-10/CIFAR-100** are datasets that consist of 10 and 100 classes, respectively, with image resolutions of $32 \times 32$ pixels. We use a ResNet-18 architecture (He et al., 2016) to train the models for 40,000 iterations with a batch size of 256, equivalent to approximately 200 epochs. Optimization is performed using SGD with a momentum of 0.9, a weight decay of 0.0002, and an initial learning rate of 0.1. A cosine annealing scheduler (Loshchilov & Hutter, 2016) is applied with a minimum learning rate of 0.0001. Data augmentation includes a 4-pixel padding crop and random horizontal flips. The experimental setup follows that of (Zheng et al., 2022).

**Tiny-ImageNet-200** is a subset of the ImageNet-1K dataset (Le & Yang, 2015), containing 200 classes. Each class consists of 500 medium-resolution images, and the spatial size of the images is $64 \times 64$ pixels. A ResNet-18 architecture (He et al., 2016) is used, and models are trained for 60 epochs with a batch size of 128. Optimization is performed using SGD with a momentum of 0.9, a weight decay of 0.0002, and an initial learning rate of 0.1. A cosine annealing scheduler (Loshchilov & Hutter, 2016) is applied with a minimum learning rate of 0.0001. Data augmentation includes an 8-pixel padding crop and random horizontal flips. The experimental setup follows that of (Hou et al., 2020).

**ImageNet-1K** is a dataset containing 1,000 classes and 1,281,167 images in total. The images are resized to a resolution of $224 \times 224$ pixels for training. ResNet-34 (He et al., 2016) is adopted as the network architecture. Models are trained for 300,000 iterations with a batch size of 256, corresponding to approximately 60 epochs. Optimization uses SGD with a momentum of 0.9, a weight decay of 0.0001, and an initial learning rate of 0.1. A cosine annealing learning rate scheduler is employed. The experimental setup follows that of (Zheng et al., 2022).

## A.2 BASELINES

### A.2.1 DATASET PRUNING

**Random** randomly selects partial data from the full dataset to form a coreset.

**Entropy** (Paul et al., 2021) quantifies sample uncertainty. Samples with higher entropy are considered to have a greater influence on model optimization. The entropy is computed using predicted probabilities at the conclusion of training.

**Forgetting** (Toneva et al., 2018) measures the frequency of forgetting events during training. Samples that are consistently remembered can be removed with minimal impact on performance.

**CCS** (Zheng et al., 2022) applies stratified sampling variations based on importance scores to enhance coreset coverage. This method can also integrate other criteria to further improve results.

**AUM** (Pleiss et al., 2020) selects samples with the highest area under the margin. The margin is defined as the probability gap between the target class and the next largest class across all training epochs. A larger AUM suggests that a sample is of higher importance.

**EL2N** (Paul et al., 2021) selects samples based on larger gradient magnitudes, which can be effectively approximated by error vector scores. For practical evaluation, only the average of the first 10 epochs' error vector scores is considered.

**TDDS** (Zhang et al., 2024) first captures each sample's contribution throughout training to integrate detailed dynamics. It then analyzes the variability of these contributions to identify well-generalized samples.

### A.2.2 COLOR QUANTIZATION

**ColorCNN** (Zhang et al., 2024) preserves critical image structures and produces a quantized version with similar class activation maps to the original image. During training, non-differentiable parts are approximated, and regularization is used to maintain similarity and prevent premature convergence. The ColorCNN is trained end-to-end with classification loss, incorporating color jittering to improve robustness.

**ColorCNN+** (Hou et al., 2024) is an extended model that supports multiple color space sizes by reducing feature dimensions from a predefined number $D$ to the desired color space size $C$ using channel-wise average pooling. Unlike the original ColorCNN, which adapts the network structure based on the color space size.

**CQFormer** (Su et al., 2023) is a model designed for color quantization using a dual-branch architecture. The annotation branch processes feature activation using a UNeXt network, while the palette branch generates color palettes through a Palette Attention Module (PAM) with reference queries. The model operates across training and testing stages, incorporating intra-cluster color similarity regularization to enhance color classification and detection capabilities by learning a compact color representation.

**MedianCut** (Heckbert, 1982) is a technique used to reduce the number of colors in an image while preserving its overall visual appearance. The algorithm works by recursively dividing the color space into smaller regions (bins) based on the distribution of pixel colors.

| | CIFAR-10 | | | | | | CIFAR-100 | | | | | | Tiny-ImageNet | | | | | |
|---|---|---|---|---|---|---|---|---|---|---|---|---|---|---|---|---|---|---|
| | Color Bits | | | | | | Color Bits | | | | | | Color Bits | | | | | |
| | 1 | 2 | 3 | 4 | 5 | 6 | 1 | 2 | 3 | 4 | 5 | 6 | 1 | 2 | 3 | 4 | 5 | 6 |
| OCTree | 57.91 | 68.95 | 75.12 | 77.24 | 80.29 | 84.99 | 27.93 | 30.15 | 39.02 | 43.34 | 47.39 | 52.89 | 24.63 | 26.89 | 34.78 | 40.12 | 44.16 | 49.45 |
| | ±0.52 | ±0.63 | ±0.71 | ±0.68 | ±0.75 | ±0.25 | ±0.42 | ±0.45 | ±0.52 | ±0.56 | ±0.61 | ±0.65 | ±0.38 | ±0.41 | ±0.48 | ±0.53 | ±0.57 | ±0.62 |
| CQFormer | 49.52 | 52.53 | 62.91 | 71.34 | 75.49 | 79.89 | 19.92 | 20.01 | 29.95 | 32.81 | 39.62 | 41.89 | 19.91 | 20.01 | 29.95 | 32.81 | 39.65 | 41.89 |
| | ±0.48 | ±0.51 | ±0.58 | ±0.65 | ±0.69 | ±0.73 | ±0.35 | ±0.36 | ±0.45 | ±0.47 | ±0.52 | ±0.54 | ±0.34 | ±0.35 | ±0.44 | ±0.46 | ±0.51 | ±0.53 |
| ColorCNN+ | 48.77 | 49.66 | 59.92 | 68.34 | 72.59 | 82.89 | 18.53 | 19.52 | 27.92 | 29.34 | 36.49 | 40.99 | 15.21 | 16.38 | 24.53 | 26.12 | 33.25 | 37.55 |
| | ±0.47 | ±0.48 | ±0.56 | ±0.63 | ±0.67 | ±0.75 | ±0.33 | ±0.34 | ±0.42 | ±0.44 | ±0.49 | ±0.52 | ±0.31 | ±0.32 | ±0.38 | ±0.41 | ±0.46 | ±0.49 |
| ColorCNN | 44.12 | 59.48 | 70.23 | 80.34 | 84.49 | 86.89 | 16.54 | 22.81 | 34.93 | 38.64 | 40.19 | 45.39 | 13.28 | 19.42 | 31.63 | 36.89 | 42.11 | |
| | ±0.45 | ±0.55 | ±0.64 | ±0.72 | ±0.77 | ±0.79 | ±0.32 | ±0.37 | ±0.48 | ±0.51 | ±0.53 | ±0.57 | ±0.29 | ±0.34 | ±0.45 | ±0.48 | ±0.49 | ±0.54 |
| MedianCut | 70.91 | 78.15 | 81.02 | 84.34 | 86.39 | 88.89 | 29.96 | 35.15 | 42.02 | 49.34 | 51.39 | 55.89 | 26.58 | 31.87 | 38.74 | 46.12 | 48.15 | 52.56 |
| | ±0.65 | ±0.71 | ±0.74 | ±0.76 | ±0.78 | ±0.81 | ±0.44 | ±0.48 | ±0.54 | ±0.61 | ±0.63 | ±0.67 | ±0.41 | ±0.46 | ±0.51 | ±0.58 | ±0.59 | ±0.64 |
| **Ours** | **79.94** | **89.15** | **91.02** | **93.34** | **94.39** | **94.89** | **38.44** | **57.69** | **65.02** | **68.09** | **70.31** | **71.55** | **36.44** | **50.51** | **52.12** | **55.09** | **57.61** | **60.02** |
| | ±0.72 | ±0.33 | ±0.84 | ±0.86 | ±0.87 | ±0.88 | ±0.51 | ±0.67 | ±0.73 | ±0.76 | ±0.78 | ±0.79 | ±0.49 | ±0.62 | ±0.64 | ±0.66 | ±0.68 | ±0.71 |

Table 6: Performance comparison for different methods across varying color bits on CIFAR-10, CIFAR-100, and Tiny-ImageNet datasets. Mean and standard deviation are computed from 5 independent runs.

**OCTree** (Gervautz & Purgathofer, 1988) builds an octree data structure where each node represents a color or a group of similar colors. Starting with all colors of an image in the octree, the process iteratively merges nodes (colors) that are closest together until a target number of colors is reached.

All images are randomly sampled from the train-set to ensure representativeness. The visualization demonstrates how our method progressively reduces the color palette while maintaining the visual fidelity of the original images. As the quantization ratio increases (i.e., fewer color bits), we observe that our approach effectively preserves key structural details and salient features, even under aggressive color reduction scenarios.

# B ADDITIONAL INFORMATION

## B.1 PRIMARY TABLE FOR COLOR QUANTIZATION

Table 6 shows the results of color quantization on different datasets. We ran each experiment five times to calculate the mean and standard deviation. The experimental results demonstrate that our color quantization algorithm outperforms other color quantization algorithms in improving the performance of the train-set after color quantization.

# C ADDITIONAL EXPERIMENTS

## C.1 ABLATION OF SOBEL OPERATOR

Table 7 demonstrates that incorporating edge distribution minimization consistently improves the color quantization performance across all bit depths (1-5) on CIFAR-10. Notably, we observe performance gains ranging from 2.26% to 3.10%, with the most significant improvement occurring

at 1-bit quantization (76.8% → 79.90%). These results validate that preserving edge information during quantization leads to better overall performance.

Table 7: Effect of using minimizing the edge distribution differences on CIFAR-10.

| Color bit | 1 | 2 | 3 | 4 | 5 |
|---|---|---|---|---|---|
| No using | 76.80 | 87.05 | 89.22 | 91.33 | 92.13 |
| **using** | **79.90** | **89.15** | **91.02** | **93.34** | **94.39** |

## C.2 Generalization to Different Networks

Table 8 presents a comparative analysis of our proposed algorithm against existing dataset pruning methods across Shufflenet (Zhang et al., 2018) and MobileNet-v2 (Sandler et al., 2018) architectures. The empirical results demonstrate that our approach consistently outperforms baseline methods across different compression ratios, achieving superior accuracy on both network architectures.

Table 8: Comparison of dataset pruning algorithms and our algorithm on CIFAR-10 with ShufffleNet and MobileNet-v2.

| | ShufffleNet, CIFAR-10 | | | | | MobileNet-v2, CIFAR-10 | | | | |
|---|---|---|---|---|---|---|---|---|---|---|
| | 80% | 83% | 87.5% | 92% | 96% | 80% | 83% | 87.5% | 92% | 96% |
| Random | 83.05 | 80.29 | 72.18 | 70.34 | 60.18 | 81.25 | 78.09 | 70.38 | 68.04 | 57.88 |
| EL2N | 69.20 | 65.32 | 23.68 | 22.13 | 20.19 | 67.20 | 63.02 | 21.88 | 20.13 | 18.19 |
| AUM | 54.64 | 45.49 | 27.78 | 20.62 | 17.35 | 55.84 | 46.59 | 27.78 | 23.30 | 19.05 |
| $CCS_{AUM}$ | 82.14 | 80.32 | 77.51 | 65.94 | 63.45 | 80.64 | 78.02 | 75.01 | 63.44 | 61.15 |
| TDDS | 83.08 | 81.65 | 80.42 | 66.55 | 62.98 | 81.28 | 79.15 | 77.92 | 65.05 | 59.48 |
| **Ours** | **89.9** | **87.98** | **86.31** | **83.67** | **74.41** | **91.19** | **88.84** | **89.25** | **85.17** | **76.26** |

Table 9 shows a comparative analysis of our proposed algorithm against existing dataset pruning methods across Swin Transformer (Liu et al., 2021) and ViT-Small (Lee et al., 2021).

Table 9: Comparison of dataset pruning algorithms and our algorithm on Transformer-Based Model. For CIFAR-10, we tested our algorithms on Swin Transformer; For CIFAR-100, we tested our algorithms on ViT-Small.

| | CIFAR-10 | | | | | CIFAR-100 | | | | |
|---|---|---|---|---|---|---|---|---|---|---|
| | 80% | 83% | 87.5% | 92% | 96% | 80% | 83% | 87.5% | 92% | 96% |
| Random | 82.26 | 79.18 | 77.09 | 69.27 | 64.18 | 33.83 | 30.96 | 26.19 | 25.61 | 22.68 |
| EL2N | 59.13 | 55.15 | 27.77 | 24.97 | 21.64 | 22.49 | 14.55 | 11.67 | 9.18 | 8.76 |
| AUM | 65.61 | 56.51 | 28.65 | 23.22 | 20.17 | 22.58 | 15.32 | 12.88 | 10.15 | 6.08 |
| $CCS_{AUM}$ | 84.38 | 80.82 | 75.18 | 71.22 | 67.01 | 59.18 | 55.35 | 51.16 | 27.13 | 22.84 |
| **Ours** | **87.66** | **84.91** | **80.82** | **76.83** | **70.54** | **68.38** | **65.19** | **63.72** | **47.19** | **35.42** |

## C.3 Ablation of Feature Extraction Algorithms

During the Cluster for Color Perceptual Consistency step, we used a pre-trained model to extract features. Table 10 compares the clustering performance of different features, including PCA, Gray Level Co-occurrence Matrix (GLCM), Color Coherence Vector (CCV), RGB Color Histograms, and Shallow-layer Feature Maps. The results show that clustering with Shallow-layer Feature Maps achieves the best performance.

Table 10: Different feature extraction strategies on CIFAR-10 trained on ResNet-18.

| Color Bits | 1 | 2 | 3 | 4 | 5 |
|---|---|---|---|---|---|
| PCA | 50.11 | 61.78 | 70.23 | 73.45 | 78.47 |
| Gray Level Co-occurrence Matrix | 70.15 | 83.17 | 85.49 | 86.17 | 90.75 |
| Color Coherence Vector | 74.15 | 86.97 | 87.19 | 89.77 | 92.35 |
| RGB Color Histograms | 75.95 | 86.18 | 88.49 | 90.17 | 91.45 |
| **Shallow-layer Feature Map** | **79.90** | **89.15** | **91.02** | **93.34** | **94.39** |

Table 11 presents the results of using different layers for feature extraction. Specifically, we evaluate Shallow-layer feature maps from the first residual block, Mid-layer1 feature maps from the second residual block, Mid-layer2 feature maps from the third residual block, and Final-layer feature maps from the final residual block. The results indicate that Shallow-layer feature maps achieve the best performance.

Table 11: Comparison of Different layers on CIFAR-10 trained on ResNet-18.

| Color Bits | 1 | 2 | 3 | 4 | 5 |
|---|---|---|---|---|---|
| Mid-layer2 feature maps | 69.15 | 79.81 | 81.09 | 83.76 | 87.75 |
| Mid-layer1 feature maps | 74.15 | 86.97 | 87.19 | 89.77 | 92.35 |
| Final-layer Feature Map | 42.10 | 53.78 | 66.39 | 75.15 | 80.44 |
| **Shallow-layer Feature Map** | **79.90** | **89.15** | **91.02** | **93.34** | **94.39** |

## C.4 ABLATION OF GRAD-CAM++ PERCENTAGE

Table 12 presents the performance under different Grad-CAM++ retention percentages. The results indicate that preserving 50% of the pixels achieves the best performance.

Table 12: Utilizing Grad-CAM++ attention maps to maintain the pixel distribution ratio $k_{Gra}\%$ on CIFAR-10 and ResNet-18.

| Color Bits | 1 | 2 | 3 | 4 | 5 |
|---|---|---|---|---|---|
| 80% | 75.95 | 87.17 | 88.11 | 89.98 | 93.05 |
| 20% | 77.31 | 87.79 | 88.23 | 90.55 | 91.47 |
| 70% | 76.15 | 86.97 | 87.41 | 90.99 | 92.55 |
| 60% | 76.85 | 87.17 | 88.89 | 91.97 | 93.01 |
| 30% | 77.95 | 88.18 | 89.19 | 92.17 | 93.45 |
| **50%** | **79.90** | **89.15** | **91.02** | **93.34** | **94.39** |

## D ADDITIONAL ANALYSIS

### D.1 EXPERIMENTS IN OTHER TASKS

Table 13a and Table 13b show that our algorithm outperforms other dataset pruning algorithms on both image segmentation and object detection tasks under high compression ratios.

Table 13: Comparison of different dataset pruning algorithms on MS COCO.

(a) Image Segmentation (MS COCO), Full dataset $AP_{50} = 55.20\%$.

| Method | 80% | 87.5% | 92% | 96% |
|---|---|---|---|---|
| Random | 45.60 | 42.07 | 38.15 | 25.99 |
| CCS | 43.68 | 39.91 | 35.58 | 26.46 |
| EL2N | 38.98 | 36.12 | 29.91 | 24.26 |
| **Ours** | **47.38** | **44.29** | **41.68** | **28.77** |

(b) Object Detection (MS COCO), Full dataset $AP_{50}^{bb} = 58.30\%$.

| Method | 80% | 87.5% | 92% | 96% |
|---|---|---|---|---|
| Random | 48.80 | 44.07 | 40.15 | 31.89 |
| CCS | 46.71 | 42.89 | 37.08 | 30.15 |
| EL2N | 40.05 | 39.15 | 31.48 | 27.36 |
| **Ours** | **50.78** | **46.29** | **42.08** | **33.37** |

Table 14 shows that our algorithm outperforms other dataset pruning algorithms on generative tasks.

Table 14: Comparison of dataset pruning algorithms on the CIFAR-10 PSLD generative task (Pandey & Mandt, 2023). Results are evaluated by FID ($\downarrow$), where lower values indicate better performance.

| Method | 80% | 87.5% | 92% | 96% |
|---|---|---|---|---|
| Random | 16.8 | 39.4 | 50.2 | 119.8 |
| CCS | 15.7 | 37.8 | 52.8 | 121.1 |
| EL2N | 18.1 | 45.1 | 61.4 | 138.3 |
| **Ours** | **11.9** | **21.3** | **42.2** | **113.5** |

## D.2 COMPARISON WITH OTHER TASKS

Table 15a, Table 15b and Table 16 shows comparison with other dataset compression algorithms like DUAL (Cho et al., 2025), DQ (Zhou et al., 2023), DQAL (Zhao et al., 2024), ADQ (Li et al., 2025) and AutoPalette (Yuan et al., 2024), our algorithms outperform than other dataset pruning algorithms.

Table 15: Comparison of our DCQ with different dataset compression algorithms.

(a) CIFAR-10 (ResNet-18)

| Method | 80% | 87.5% | 92% | 96% |
|---|---|---|---|---|
| ADQ | 90.40 | 87.51 | 86.32 | 75.99 |
| DQAL | 90.20 | 87.61 | 85.30 | 75.80 |
| DQ | 89.40 | 87.70 | 84.25 | 77.92 |
| DUAL | 91.42 | 88.99 | 86.15 | 74.83 |
| **Ours** | **94.39** | **91.02** | **89.15** | **79.90** |

(b) ImageNet-1k (ResNet-34)

| Method | 80% | 87.5% | 92% | 96% |
|---|---|---|---|---|
| ADQ | 65.31 | 60.91 | 47.02 | 32.19 |
| DQAL | 62.21 | 60.41 | 46.73 | 31.89 |
| DQ | 64.30 | 60.45 | 47.17 | 32.27 |
| DUAL | 66.50 | 61.15 | 46.55 | 33.83 |
| **Ours** | **66.99** | **62.02** | **49.69** | **35.95** |

Table 16: Comparison between our algorithm and AutoPalette on CIFAR-10 and CIFAR-100 (ConvNetD3)

| | CIFAR-10 | | CIFAR-100 | |
|---|---|---|---|---|
| | IPC=10 | IPC=50 | IPC=10 | IPC=50 |
| AutoPalette | 74.3 | 79.4 | 52.6 | 53.3 |
| **Ours** | **77.9** | **82.3** | **58.9** | **61.7** |

Table 17 shows that standard data augmentation further improves performance when training on our color-quantized datasets.

Table 17: Combining other dataset augmentation algorithms on CIFAR-10 (ResNet-18) under 1-bit quantization.

| Method | 80% | 87.5% | 92% | 96% |
|---|---|---|---|---|
| Original setting | 94.39 | 91.02 | 89.15 | 79.90 |
| Original setting + ColorJitter | 95.02 | 92.44 | 90.45 | 80.95 |
| Original setting + GaussianBlur | 95.47 | 92.95 | 91.47 | 81.75 |

Table 18 shows that our algorithm outperforms other algorithms on fine-grained recognition tasks.

Table 18: Different dataset pruning algorithms on a fine-grained recognition task (Stanford Cars, API-Net (Zhuang et al., 2020)). Full dataset performance = 95.3%.

| Method | 80% | 87.5% | 92% | 96% |
|---|---|---|---|---|
| Random | 85.95 | 71.28 | 68.19 | 64.55 |
| CCS | 88.98 | 74.17 | 70.07 | 67.17 |
| EL2N | 67.32 | 23.98 | 16.31 | 14.85 |
| **Ours** | **91.16** | **76.22** | **72.34** | **70.17** |

Table 19 shows that our algorithm has high robustness under label noise injection.

Table 19: Evaluation of dataset pruning algorithms on CIFAR-10 (ResNet-18) with 30% symmetric random label noise. Full dataset accuracy = 83.06%.

| Method | 80% | 87.5% | 92% | 96% |
|---|---|---|---|---|
| Random | 76.51 | 67.18 | 62.05 | 49.36 |
| CCS | 78.51 | 69.35 | 64.16 | 51.28 |
| TDDS | 77.44 | 67.99 | 63.66 | 50.85 |
| **Ours** | **82.26** | **73.44** | **68.34** | **59.17** |

Table 20 shows that even when using substantially fewer epochs (e.g., 50 epochs = 9,750 iterations), our method still outperforms SOTA pruning methods trained for 40,000 iterations, demonstrating its strong performance robustness.

Table 20: Different number of training epochs on CIFAR-10 (ResNet-18) under 1-bit color quantization

| Number of Epochs | Number of Training Iterations | Acc (%) |
|---|---|---|
| 50 | 9,750 | 75.01 |
| 100 | 19,500 | 76.19 |
| 125 | 24,375 | 77.69 |
| 150 | 29,250 | 78.57 |
| 200 | 39,000 | 79.90 |

## D.3 COMPUTATIONAL COMPLEXITY

Table 21 reports the computational complexity of DCQ, which scales linearly with the size of the dataset. Table 22 reports the time consumption of each stage in our pipeline.

Table 21: Dataset statistics including class count, image distribution, total size, and complexity.

| Dataset | #Classes (C) | Images per Class (IPC) | Total Images | Complexity |
|---------|--------------|------------------------|--------------|------------|
| ImageNet-1K | 1,000 | ≈1200 | ≈1.2M | $O(1,000 \times 1200 \times \text{BackboneFLOPs})$ |
| ImageNet-21K | 21,000 | ≈700 | ≈14.7M | $O(21,000 \times 700 \times \text{BackboneFLOPs})$ |

Table 22: Time cost of generating different datasets with color quantization (sufficient memory).

| Dataset | Palette-Learning | Attention-Computation | Differentiable-Refinement | Total Time |
|---------|------------------|-----------------------|---------------------------|------------|
| ImageNet-1K | 108 mins | 16 mins | 30 mins | 154 mins |
| CIFAR-10/100 | 20 seconds | 2 seconds | 30 seconds | 52 seconds |

# E  VISUALIZATION

Figure 7 and 8 presents the visualizations of CIFAR-10/100 and Tiny-ImageNet under different color quantization rates, respectively.

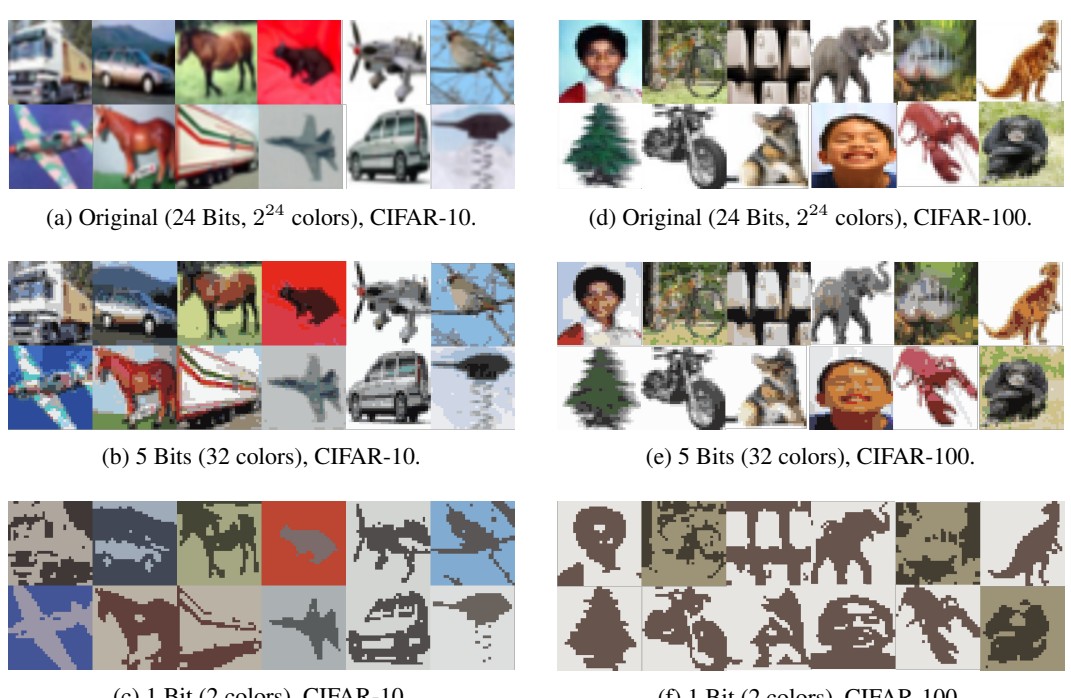

(a) Original (24 Bits, $2^{24}$ colors), CIFAR-10.  (d) Original (24 Bits, $2^{24}$ colors), CIFAR-100.

(b) 5 Bits (32 colors), CIFAR-10.  (e) 5 Bits (32 colors), CIFAR-100.

(c) 1 Bit (2 colors), CIFAR-10.  (f) 1 Bit (2 colors), CIFAR-100.

Figure 7: Visualization of different datasets under different color quantization ratios. Left column: CIFAR-10 images; Right column: CIFAR-100 images. From top to bottom: original images, 5-bits quantization (32 colors), and 1-bit quantization (2 colors).

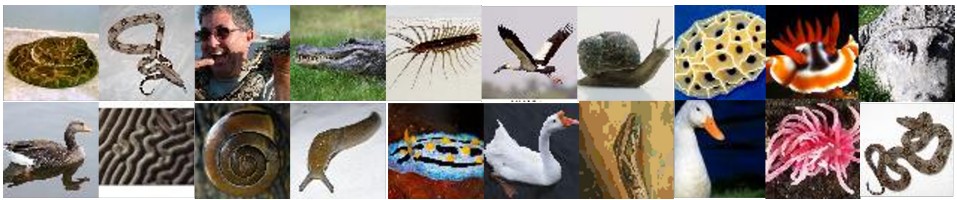

(a) Original (24 Bits, $2^{24}$ colors), Tiny-ImageNet.

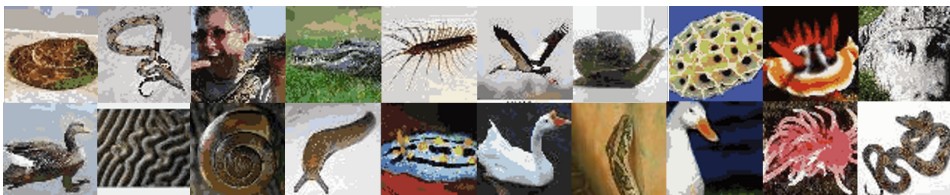

(b) 5 Bit (32 colors), Tiny-ImageNet

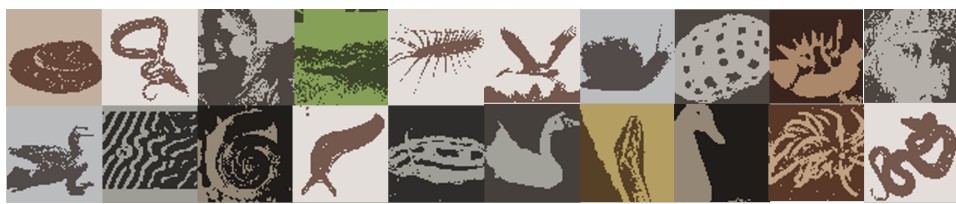

(c) 1 Bit (2 colors), Tiny-ImageNet

Figure 8: Visualization of Tiny-ImageNet under different color quantization ratios. From top to bottom: original images, 5-bits quantization (32 colors), and 1-bit quantization (2 colors).

Figure 9 and 10 present the visualizations of CIFAR-10/100 and Tiny-ImageNet under different color quantization rates, respectively.

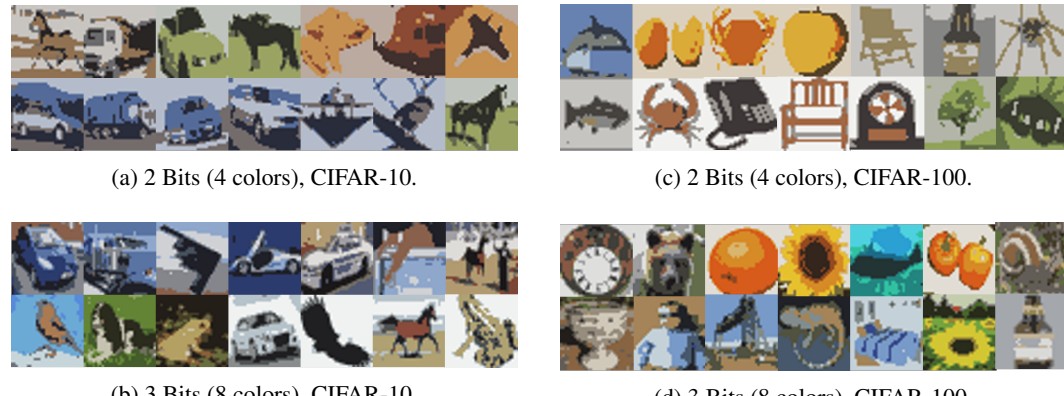

(a) 2 Bits (4 colors), CIFAR-10.

(c) 2 Bits (4 colors), CIFAR-100.

(b) 3 Bits (8 colors), CIFAR-10.

(d) 3 Bits (8 colors), CIFAR-100.

Figure 9: Visualization of different datasets under different color quantization ratios. Left column: CIFAR-10 images; Right column: CIFAR-100 images. From top to bottom: 2-bit quantization (4 colors), and 3-bit quantization (8 colors).

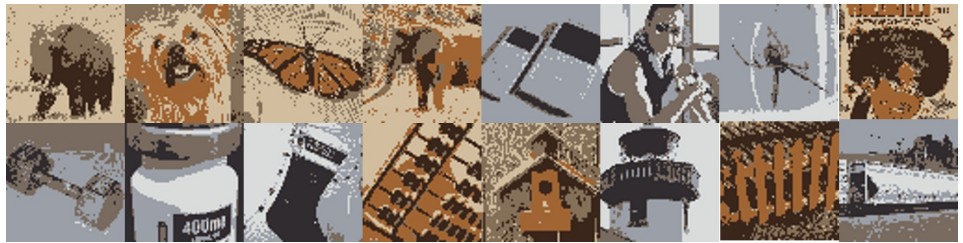

(a) Original (2 Bit (4 colors), Tiny-ImageNet.

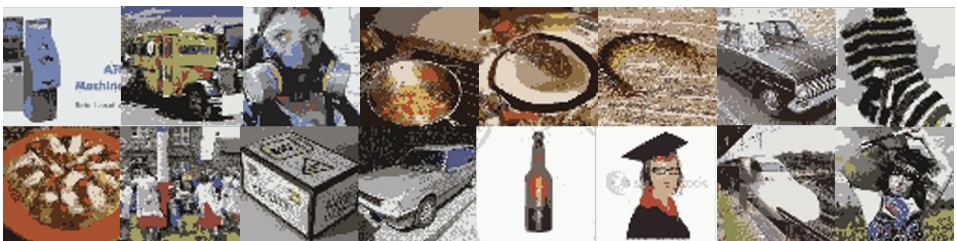

(b) 3 Bit (8 colors), Tiny-ImageNet

Figure 10: Visualization of Tiny-ImageNet under different color quantization ratios. From top to bottom: 2-bit quantization (4 colors) and 3-bit quantization (8 colors).

