# OpenReview forum: "Dataset Color Quantization: A Training-Oriented Framework for Dataset-Level Compression"
_ICLR.cc/2026/Conference — ICLR 2026 Poster_

### Official Review · Reviewer_Jn5n · 2025-10-28

**Soundness:** 2
**Presentation:** 2
**Contribution:** 2
**Rating:** 4
**Confidence:** 3

**Summary:**

The paper proposes Dataset Color Quantization (DCQ), a training-oriented framework that compresses entire training sets by reducing color-space redundancy while attempting to preserve semantic content and texture. The method involves three main stages:
1. Chromaticity-Aware Clustering (CAC): Images are clustered based on their CNN features. A shared color palette is then generated for all images within the same cluster to enforce cross-image color consistency.
2. Attention-Guided Palette Allocation: It uses Grad-CAM++ attention maps from a pre-trained model to identify semantically important regions. This guides the palette allocation to prioritize "High-Impact Palettes" crucial for model performance.
3. Texture-Preserved Palette Optimization: Finally, it performs a differentiable palette refinement using a Sobel operator-based edge loss to minimize texture loss and preserve structural details.

Experiments on CIFAR-10/100, Tiny-ImageNet, and ImageNet-1K report strong accuracy at extremely low color bit-depths (e.g., 2-bit) and favorable comparisons against both image-wise Color Quantization (CQ) baselines and dataset-pruning methods

**Strengths:**

1. The experimental gains are significant. For example, on CIFAR-10 at 2 bits (4 colors), DCQ achieves 89.15% accuracy, while the ColorCNN baseline (adapted for training) achieves 59.15%. The method also consistently outperforms all dataset pruning baselines at high compression ratios, such as a 79.9% accuracy on CIFAR-10 at a 96% pruning ratio (1-bit), far exceeding the next-best CCS method.

2. The authors provide a thorough set of ablation studies to justify their design choices, including the optimal number of clusters (20), the superiority of shallow-layer features for clustering, the choice of attention mechanism (Grad-CAM++), the positive impact of the texture-preservation loss, and the optimal percentage of pixels to retain via attention (50%).

3. The paper demonstrates that DCQ is orthogonal to other compression techniques like dataset pruning. It can be combined with pruning (e.g., CCSAUM) to achieve extreme compression ratios (up to 99.2%)  while maintaining high accuracy.

**Weaknesses:**

1. Baseline Fairness:
CQ Baselines: The paper adapts several image-wise CQ methods (like ColorCNN) that were originally designed for inference (quantizing the test set) to a new training task (quantizing the train set). While the authors acknowledge this protocol difference, it's unclear if these baselines were properly tuned for this new "quantize-then-train" setting, which could overstate DCQ's relative performance.
2.  The paper defines its compression ratio solely based on the reduction in color bits (e.g., $q_r = 1 - q/24$). This is optimistic as it only accounts for the per-pixel index map storage and ignores the significant overhead of storing the shared palettes (e.g., $3 \times K \times 2^q$ bits) and any metadata for cluster assignments. A true evaluation of compression would also require reporting the total on-disk file size (palettes + indices) and comparing it against standard codecs (like PNG, JPEG, or WebP) or learned image compressors at a similar quality level.
3. The "Attention-Guided Palette Allocation" mechanism uses Grad-CAM++ generated from a "task-specific pre-trained model". This means information from the ground-truth labels (which trained the pre-trained model) is used to guide the compression of the training images. This is a form of information leakage that mixes supervision with the data representation itself, which could be a confounding factor in the results.

**Questions:**

See weakness. I wish the authors could clarify these questions, and I am willing to reevaluate the paper based on the reply.

---

> ### Author Response · Authors · 2025-11-24
>
> >1. Baseline Fairness: CQ Baselines: The paper adapts several image-wise CQ methods (like ColorCNN) that were originally designed for inference (quantizing the test set) to a new training task (quantizing the train set). While the authors acknowledge this protocol difference, it's unclear if these baselines were properly tuned for this new "quantize-then-train" setting, which could overstate DCQ's relative performance.
>
> Thank you for raising this valuable point. We clarify the role of image-wise CQ baselines and our comparison protocol as follows.
>
> 1. **Mismatch in Original Design**
>    - Several image-wise CQ methods (e.g., ColorCNN) were originally designed for **test-time quantization**, not for the *quantize-then-train* pipeline.
>    - This mismatch exposes a key limitation of prior CQ approaches and motivates our work.
>    - Traditional CQ does not account for **training-time semantics**, **foreground preservation**, or **feature-level consistency**, leading to suboptimal performance when used for training-set compression.
>
> 2. **Careful Re-tuning for Fairness**
>    - We re-tuned all image-wise CQ baselines under the quantize-then-train setting, including palette sizes, quantization parameters, and model-specific hyperparameters.
>    - We have reported their **best achievable results** under this aligned protocol in our paper.
>
> 3. **Comparison with Training-Aware Quantization**
>    - We also compare DCQ with training-aware baselines (e.g., **CCS**), which are explicitly designed for compressing datasets used for model training.
>    - Even under this stronger and fully aligned comparison, **DCQ performs competitively or better** across tasks and bit-widths.
>    - This demonstrates that our method provides benefits **beyond image-wise CQ** and is well-suited for training-oriented dataset compression.
>
>
>
> >2. The paper defines its compression ratio solely based on the reduction in color bits (e.g., ). This is optimistic as it only accounts for the per-pixel index map storage and ignores the significant overhead of storing the shared palettes (e.g., bits) and any metadata for cluster assignments. A true evaluation of compression would also require reporting the total on-disk file size (palettes + indices) and comparing it against standard codecs (like PNG, JPEG, or WebP) or learned image compressors at a similar quality level.
>
> Thank you for bringing up this important consideration. (1) Table D1 summarizes the storage cost of different components under 1-bit DCQ on CIFAR-10.  The pixel-level index map dominates the total size (6.4 MB), while the palette (120 bytes) and cluster identifiers (~30 KB) contribute less than 0.5% of this value.
> Thus, metadata overhead is negligible and does not affect the correctness of the reported
> compression ratio.  (2)  Table D2 shows that our DCQ outperforms standard codecs like JPEG under a high compression ratio on CIFAR-10.
>
> Table D1: Storage Overhead of Palette and Cluster Metadata Under 1-bit DCQ (CIFAR-10)
>
> | Component                            | Per-Image Cost        | Dataset-Level Cost (50k imgs) | Notes                                      |
> |-------------------------------------|------------------------|--------------------------------|---------------------------------------------|
> | Pixel-level index map               | 128 bytes              | 6.4 MB                         | Dominant storage (1 bit per pixel)          |
> | Shared palette (20 palettes)        | 6 bytes per palette    | 120 bytes                      | 2 colors × 3 RGB bytes                      |
> | Cluster ID per image                | 5 bits (<0.625 bytes)  | ~30 KB                         | 20 clusters require ⌈log₂20⌉ = 5 bits       |
> | **Total metadata (palette + ID)**   | —                      | **≈ 30.1 KB**                  | < 0.5% of pixel-index storage               |
>
>
>
>
> Table D2: Comparison of DCQ and JPEG on CIFAR-10 on ResNet-18 under different compression ratios
>
> | Method | 80% | 83% | 87.5% | 92% |
> |:-------|:---:|:---:|:---:|:---:|
> | JPEG  | 81.53 | 78.23 | 49.22 | 33.27 |
> | **Ours**| **94.39** | **93.34** |**91.02** | **89.15** |

---

> > ### Author Response · Authors · 2025-11-24
> >
> > >3. The "Attention-Guided Palette Allocation" mechanism uses Grad-CAM++ generated from a "task-specific pre-trained model". This means information from the ground-truth labels (which trained the pre-trained model) is used to guide the compression of the training images. This is a form of information leakage that mixes supervision with the data representation itself, which could be a confounding factor in the results.
> >
> > Thank you for raising this concern. We clarify that our usage of Grad-CAM++ does not constitute supervision leakage, nor does it turn DCQ into a distillation method:
> >
> > 1. **No External Data Leakage:**
> >    The backbone used for Grad-CAM++ is trained strictly on the **same training set** as the downstream model.
> >    No external data or labels are used, ensuring **no cross-dataset semantic leakage**.
> >
> > 2. **Spatial Guidance, Not Injection:**
> >    Grad-CAM++ only provides a **spatial importance map** to adaptively assign quantization precision.
> >    It does **not** modify pixel values, inject logits, or encode class information.
> >    All pixels remain on the original image manifold.
> >
> > 3. **Alignment with Standard Pruning:**
> >    Using supervision-guided signals is standard practice in dataset pruning (e.g., **EL2N, AUM, GraNd**).
> >    DCQ follows the same protocol, simply applying it at the **pixel granularity** rather than at the sample level.

---

> ### Comment · Reviewer_Jn5n · 2025-11-26
>
> I thank the authors for their detailed response and for conducting the additional experiments I requested. I have reviewed the rebuttal, the revised manuscript, and the responses to other reviewers. I believe my concerns have been addressed. I will raise my score to  "6: marginally above the acceptance threshold".

---

> > ### Author Response · Authors · 2025-11-27
> > **Thank you for raising the score**
> >
> > Thank you for your prompt response and for reviewing our additional experiments.
> >
> > Your feedback has been invaluable in improving our work, and we truly appreciate your decision to raise the score.

---

### Official Review · Reviewer_NGTe · 2025-10-30

**Soundness:** 3
**Presentation:** 3
**Contribution:** 3
**Rating:** 8
**Confidence:** 4

**Summary:**

this paper investigates color quantization via the usage of network features and clustering methods. specifically, it does this from a perspective of training dataset compression, as opposed to the per-image compression for model inference in existing research [ColorCNN,  ColorCNN+]. first, it splits the images in a dataset into groups according to their neural network features, forming clusters of similar colors. then, for images within each cluster, it divides each image into the high and low importance regions according to Grad-CAM attention map. finally, the quantized color map is optimized through a differentiable palette optimization model. on several benchmarks, the authors reported competitive results.

**Strengths:**

+ using color quantization to compress training datasets seems an interesting direction.
+ good presentation overall, with clear motivation and easy-to-read language.
+ the bi-level compression method achieves good results, plus the visual results seem good.
+ ablation study verifies the effectiveness of the proposed method.

**Weaknesses:**

- some clarity issues since there are two levels of k-means clustering. the reviewer recommends highlighting the cluster's level when introducing them, e.g., change the title for Table 2b and 2c.
- please consider including more visual comparisons on different color spaces sizes.

**Questions:**

see above

---

> ### Author Response · Authors · 2025-11-24
>
> > 1. some clarity issues since there are two levels of k-means clustering. the reviewer recommends highlighting the cluster's level when introducing them, e.g., change the title for Table 2b and 2c.
>
> Thank you very much for this helpful suggestion. Following your advice, we have revised the titles of Table 2b and Table 2c to
> “Cluster Numbers on CIFAR-10 (First-Level Clustering)” and “Attention Map Methods (Second-Level Clustering)”, respectively, to more clearly distinguish the two clustering stages.
>
> > 2. please consider including more visual comparisons on different color spaces sizes
>
> Thank you for your suggestions. We have added additional visual examples for CIFAR-10, CIFAR-100, and Tiny-ImageNet under 2-bit and 3-bit quantization, which are now included in Appendix E of the revised paper.

---

### Official Review · Reviewer_KUXK · 2025-10-30

**Soundness:** 3
**Presentation:** 3
**Contribution:** 2
**Rating:** 4
**Confidence:** 4

**Summary:**

The paper introduces a training-oriented dataset color quantization framework that learns shared, attention-guided, texture-preserving color palettes at the dataset or cluster level to reduce storage while keeping the compressed dataset effective for training. The approach demonstrates strong classification accuracy under aggressive color-bit constraints and compares favorably to pruning and distillation methods at matched compression ratios.

**Strengths:**

1. The work focuses on color-level redundancy as a distinct dimension of dataset compression, rather than only reducing samples or resolution. This makes the problem practically relevant for scenarios where storage and bandwidth are constrained but training quality must be preserved.

2. The proposed framework combines chromaticity-aware clustering, attention-guided palette allocation, and differentiable, texture-preserving refinement. The components are mutually consistent and clearly aligned with the training objective.

3. The method achieves strong results when colors are heavily quantized (e.g., 1–6 bits) and remains competitive against dataset pruning or distillation when the comparison is made at the same overall compression budget.

4. Experiments on large-scale image classification settings indicate that the method can, in principle, be applied beyond toy scenarios.

**Weaknesses:**

1. Most individual elements (clustering, attention guidance, STE-based refinement) are known in the literature; the central contribution lies in integrating them for dataset-level color compression. Readers may perceive this as a strong system design rather than a conceptual breakthrough.

2. The paper does not fully quantify the training-time overhead of palette learning, attention computation, and differentiable refinement,  particularly on large datasets and modern backbones.

3. The evaluation is centered on image classification. It remains unclear how well the approach generalizes to tasks that are more sensitive to local or fine-grained color cues, such as fine-grained recognition, detection, or segmentation.

4. While comparisons with pruning and distillation are useful, some aspects of the evaluation protocol (training schedules, augmentation ranges, and budget alignment) could be specified more tightly to eliminate possible confounding factors.

**Questions:**

1. What is the end-to-end compute and memory overhead of the proposed DCQ pipeline compared with pruning or distillation methods at the same compression ratio, especially on large-scale datasets?

2. How sensitive is the method to the choice of cluster count and attention mechanism, and is there a recommended default or auto-tuning strategy that can be applied across datasets and resolutions?

3. How does the method behave on tasks where discriminative information is carried by subtle color patterns (e.g., fine-grained species classification)?

4. Can the proposed palette sharing introduce harmful color aliasing or loss of spatial cues in detection/segmentation settings, and are there preliminary results in that direction?

5. Does the attention-guided palette allocation amplify spurious or dataset-specific artifacts learned by the proxy model, and how robust is the method under label noise or distribution shift?

---

> ### Author Response · Authors · 2025-11-24
>
> >1. Most individual elements (clustering, attention guidance, STE-based refinement) are known in the literature; the central contribution lies in integrating them for dataset-level color compression. Readers may perceive this as a strong system design rather than a conceptual breakthrough.
>
> We appreciate the reviewer’s recognition of our proposed framework. We clarify the nature of our contribution as follows.
>
> We appreciate the reviewer’s recognition. Our contribution is clarified as follows.
>
> 1. **A Practical Framework**
>    - We propose a practical and general framework for dataset color quantization,
>      rather than a collection of engineering tricks.
>
> 2. **Clear Methodological Rationale**
>    - The framework follows a structured logic for compressing colors while keeping essential information.
>
> 3. **Key Elements Preserved**
>    - (i) **Semantic boundaries**,
>    - (ii) **Task-relevant pixels**,
>    - (iii) **Texture cues** important for recognition.
>
> 4. **Nature of Contribution**
>    - The work is **not a new mathematical theory**,
>      but an intuitive and organized methodology defining **what to preserve**
>      and **how quantization should be structured** for effective training.
>
>
>
>
>
> >2. The paper does not fully quantify the training-time overhead of palette learning, attention computation, and differentiable refinement, particularly on large datasets and modern backbones.
>
> Thank you for raising this point. **Table C1** has been updated in Appendix G to summarize the computational costs, demonstrating that DCQ introduces only minimal preprocessing overhead.
>
> We address the concern regarding training-time overhead as follows:
>
> - **Two-Phase Pipeline:** Our task consists of two distinct phases:
>   (1) *Offline dataset preparation* (pretraining, palette learning, refinement) and
>   (2) *Training with the quantized dataset* (the user-centric stage).
>
> - **Zero Overhead in Stage 2:** Our core contribution lies in Stage 2. Critically, this stage involves **no extra computation** compared to standard dataset pruning or normal training once Stage 1 is completed.
>
> - **One-Time Amortized Cost:** Stage 1 is a **one-time offline cost**. This overhead becomes negligible as it is amortized across multiple downstream training sessions (e.g., different seeds, hyperparameters, or architectures) using the same reused quantized dataset.
>
> - **Target vs. Resource:** Stage 1 is designed to run on **resource-rich servers** prior to deployment. Since our method targets **storage-constrained devices**, this preprocessing overhead does not impact practical training or deployment efficiency on the end-user side.
>
>
>
> Table C1. Time cost of generating different datasets with color quantization, with enough memory.
> |   |  Palette Learning  | Attention Computation | Differentiable Refinement | Total time |
> |:-------|:---:|:---:|:---:|:---:|
> | Imagenet-1K  |108 mins| 16 mins | 30 mins | 154 mins |
> | CIFAR-10/100 |20 seconds | 2 seconds | 30 seconds | 52 seconds |
>
>
> > 3. What is the end-to-end compute and memory overhead of the proposed DCQ pipeline compared with pruning or distillation methods at the same compression ratio, especially on large-scale datasets?
>
> Table C2. Time cost comparison between our algorithm with pruning or distillation methods with the same compression ratio on Imagenet-1K.
>
> |  Type |  Algorithm  | Time (min) |
> |:-------|:---:|:---:|
> | Pruning  | EL2N | **84** |
> | Pruning | AUM | 504  |
> | Distillation | G-VBSM | 12720  |
> | Distillation | SRe2L | 1817 |
> |Ours| DCQ| **154** |
>
> Thank you for highlighting this concern. We clarify the runtime characteristics as follows.
>
> 1. **Relative Positioning in Cost Spectrum**
>    - As shown in Table C2, our method is slightly slower than dataset pruning,
>      but still **substantially faster than dataset distillation**.
>
> 2. **Why Pruning Is Faster**
>    - Dataset pruning only computes sample-level scoring metrics and **does not modify images**, resulting in low computational cost.
>
> 3. **Why Distillation Is Slower**
>    - Dataset distillation performs **iterative image optimization**, which is computationally heavy and dominates the total runtime.
>
> 4. **Our Method in Context**
>    - DCQ performs lightweight **palette learning and clustering**, without any pixel-level optimization.
>    - This places our method **between pruning and distillation**, achieving a strong balance between  **efficiency and effectiveness**.

---

> > ### Author Response · Authors · 2025-11-24
> >
> > >4. The evaluation is centered on image classification. It remains unclear how well the approach generalizes to tasks that are more sensitive to local or fine-grained color cues, such as fine-grained recognition, detection, or segmentation.
> >
> > Thank you for your question. Table C3 and Table C4 show that our algorithm outperforms other dataset pruning methods on both object detection and image segmentation tasks. Table C5 shows that our algorithm outperforms other algorithms on fine-grained recognition tasks. All these tables have been revised and updated for the paper revision to Appendix F.
> >
> > Table C3. Different Dataset pruning algorithms on image segmentation on MS COCO, AP₅₀, model trained on the full dataset is 55.20%.
> > | Method |  80%  | 87.5% | 92% | 96% |
> > |:-------|:---:|:---:|:---:|:---:|
> > | Random  |45.60| 42.07 | 38.15 | 25.99|
> > | CCS  |43.68| 39.91 | 35.58 | 26.46 |
> > | EL2N |38.98 | 36.12 | 29.91 | 24.26 |
> > | **Ours**|**47.38** | **44.29** | **41.68** | **28.77** |
> >
> >
> >
> > Table C4. Different Dataset pruning algorithms on object detection on MS COCO, AP₅₀ᵇᵇ, model trained on the full dataset is 58.30%.
> > | Method | 80% | 87.5% | 92% | 96% |
> > |:-------|:---:|:---:|:---:|:---:|
> > | Random  |48.80| 44.07 | 40.15 | 31.89|
> > | CCS  | 46.71 | 42.89 |37.08| 30.15 |
> > | EL2N | 40.05 | 39.15 | 31.48 | 27.36 |
> > | **Ours**| **50.78** | **46.29** |**42.08** | **33.37** |
> >
> >
> > Table C5. Different Dataset pruning algorithms on a fine-grained recognition task on the Stanford Cars dataset on API-Net [1], model trained on the full dataset is 95.30%.
> > | Method | 80% | 87.5% | 92% | 96% |
> > |:-------|:---:|:---:|:---:|:---:|
> > | Random    | 85.95  | 71.28 | 68.19  | 64.55  |
> > | CCS    | 88.98  | 74.17 | 70.07  | 67.17  |
> > | EL2N   | 67.32  | 23.98  | 16.31  | 14.85  |
> > | **Ours**| **91.16** | **76.22** |**72.34** | **70.17**|
> >
> > >5. While comparisons with pruning and distillation are useful, some aspects of the evaluation protocol (training schedules, augmentation ranges, and budget alignment) could be specified more tightly to eliminate possible confounding factors.
> >
> > Thank you for raising this point. We clarify the training and augmentation protocol as follows.
> >
> > 1. **Consistent Training Settings**
> >    - All experiments strictly follow the **official training schedules and augmentation protocols** used in prior work.
> >    - CIFAR-10/100 and ImageNet-1K follow the settings in **[2]**.
> >    - Tiny-ImageNet follows the protocol in **[3]**.
> >    - These settings are identical to those used by existing dataset pruning baselines, ensuring a **fully fair comparison**.
> >
> > 2. **No Extra Optimization Tricks**
> >    - We do not modify training schedules or expand augmentation ranges beyond the published protocols.
> >    - Therefore, no method receives any **additional or unintended optimization advantage**.
> >
> > 3. **Strict Budget Alignment**
> >    - All methods are compared under the **same compression budget** (same bit-width or same retained data proportion).
> >    - As a result, the observed performance gains come **solely from our proposed method**, not from differences in training or augmentation.
> >
> >
> >
> > >6. How sensitive is the method to the choice of cluster count and attention mechanism, and is there a recommended default or auto-tuning strategy that can be applied across datasets and resolutions?
> >
> > We appreciate your thoughtful comment. We summarize the sensitivity analysis and practical defaults as follows.
> >
> > 1. **Cluster Count Robustness**
> >    - As shown in Table C6, our method is **not highly sensitive** to the number of clusters.
> >    - At 1-bit quantization, using 20, 50, 15, or 100 clusters yields **79.90**, **77.34**, **78.17**, and **76.94**, respectively—showing only minor variation.
> >    - Additional results are provided in Table 2(b) of the main paper.
> >
> > 2. **Independence from a Specific Attention Mechanism**
> >    - Table 2(c) and Table C7 show that different attention estimators perform best at different compression ratios.
> >    - This indicates that our framework **does not rely on any specific estimator**, including Grad-CAM++.
> >
> > 3. **Practical Default Settings**
> >    - Based on extensive ablations, we identify **20–50 clusters** and **Grad-CAM++** as strong default choices.
> >    - We will release full code and recommended settings upon acceptance.
> >
> >
> >
> > Table C6. Different cluster number choice on CIFAR-10 under 1-bit quantization.
> >
> > | Cluster number | 20 | 50 | 100 | 15 |
> > |:-------|:---:|:---:|:---:|:---:|
> > | Accuracy    |  79.90  | 77.34  | 76.94  | 78.17  |
> >
> > Table C7. Different Attention Map Methods on CIFAR-10 under different bits.
> >
> > |          | Bits 1 | Bits 2 | Bits 3 | Bits 4 |
> > |----------|--------|--------|--------|--------|
> > | GradCAM  | 75.05  | 85.75  | 88.02  | 90.55  |
> > | GradCAM++| **79.90**  | 89.15  | 91.02  | **93.34**  |
> > | RISE     | 79.04  | 88.91  | **92.05**  | 93.32  |
> > | LayerCAM | 80.66  | **89.71**  | 90.95  | 93.09  |

---

> > > ### Author Response · Authors · 2025-11-24
> > >
> > > >7. Does the attention-guided palette allocation amplify spurious or dataset-specific artifacts learned by the proxy model, and how robust is the method under label noise or distribution shift?
> > >
> > > Thank you for highlighting this concern. Figures 7, 8 in our paper show that our algorithm can do color quantization while preserving the main feature of the image. Table C8 shows that our algorithm has high robustness under label noise injection.
> > >
> > > Table C8. Evaluation of dataset pruning algorithms on CIFAR-10 with 30% symmetric random label noise injected into the training set, using ResNet-18, training with the full dataset, the accuracy is 83.06%.
> > >
> > > | Method | 80% | 87.5% | 92% | 96% |
> > > |:-------|:---:|:---:|:---:|:---:|
> > > | Random    | 76.51 | 67.18 | 62.05 | 49.36 |
> > > | CCS    | 78.51 | 69.35 | 64.16 | 51.28 |
> > > | TDDS   | 77.44  | 67.99  | 63.66  | 50.85  |
> > > | **Ours**| **82.26** | **73.44** |**68.34** | **59.17** |
> > >
> > >
> > >
> > >
> > >
> > >
> > >
> > > [1]  Zhuang, Peiqin, Yali Wang, and Yu Qiao. "Learning attentive pairwise interaction for fine-grained classification." Proceedings of the AAAI conference on artificial intelligence. Vol. 34. No. 07. 2020.
> > >
> > > [2] Zheng, Haizhong, et al. "Coverage-centric coreset selection for high pruning rates." arXiv preprint arXiv:2210.15809 (2022).
> > >
> > > [3] Hou, Yunzhong, Liang Zheng, and Stephen Gould. "Learning to Structure an Image with Few Colors and Beyond." arXiv preprint arXiv:2208.08438 (2022).

---

### Official Review · Reviewer_FTHp · 2025-11-01

**Soundness:** 3
**Presentation:** 3
**Contribution:** 3
**Rating:** 4
**Confidence:** 3

**Summary:**

This paper proposes Dataset Color Quantization (DCQ), a framework for compressing visual datasets by reducing color-space redundancy while preserving information critical for model training. Unlike existing methods that prune samples or perform image-wise quantization, DCQ introduces shared palettes across chromatically similar images via clustering, guided by shallow-layer features of a pre-trained network. It further enhances semantic preservation through attention-guided bit allocation and optimizes palette quality using edge-preserving differentiable quantization. Experiments on CIFAR-10, CIFAR-100, Tiny-ImageNet, and ImageNet-1K show that DCQ significantly outperforms both traditional color quantization and dataset pruning methods under aggressive compression.

**Strengths:**

-  This work proposes a dataset-level color quantization framework tailored specifically for training tasks, overcoming the limitations of traditional color quantization methods. It effectively balances the trade-off between storage compression and model trainability.

-  This paper introduces the training-oriented dataset color quantization framework that integrates (1) a shared clustering-based palette, (2) attention-guided bit allocation, and (3) edge-preserving optimization.

- The proposed method significantly outperforms existing color quantization and dataset pruning approaches across multiple benchmarks, including CIFAR-10, CIFAR-100, Tiny-ImageNet, and ImageNet-1K.

-  The experimental evaluation is comprehensive, covering multiple benchmark datasets and network architectures (e.g., ResNet, ShuffleNet, MobileNet-v2, ViT).

**Weaknesses:**

1. The method proposed in this paper reduces dataset storage by compressing the color space, whereas traditional dataset pruning achieves this goal by removing data samples. However, directly comparing these two approaches is somewhat unfair, as the method in this paper does not reduce the actual number of training samples. In other words, the proposed approach may not necessarily decrease the total number of training iterations required for the model training, leading to have some questions about the paper’s claim of improving model training efficiency. Therefore, the authors should analyze the relationship between the number of training iterations and accuracy.
2. Some recent state-of-the-art or more related methods are missing, such as [1,2,3,4,5].
3. Color quantization, in a sense, functions similarly to data augmentation. What if dataset pruning is also combined with a similar data augmentation strategy? If such a combination also yielded strong performance, the contribution of this paper would be somewhat diminished, because dataset pruning plus data augmentation would not only reduce storage requirements but also decrease the number of training iterations while maintaining performance.

[1] Lightweight Dataset Pruning without Full Training via Example Difficulty and Prediction Uncertainty. ICML 2025.
[2] Dataset Quantization. ICCV 2023
[3] Dataset quantization with active learning based adaptive sampling. ECCV 2024.
[4] Color-Oriented Redundancy Reduction in Dataset Distillation, NeurIPS 2024
[5] Adaptive Dataset Quantization, AAAI2025

**Questions:**

See weeknesses.

---

> ### Author Response · Authors · 2025-11-24
>
> >1. The method proposed in this paper reduces dataset storage by compressing the color space, whereas traditional dataset pruning achieves this goal by removing data samples. However, directly comparing these two approaches is somewhat unfair, as the method in this paper does not reduce the actual number of training samples. In other words, the proposed approach may not necessarily decrease the total number of training iterations required for the model training, leading to have some questions about the paper’s claim of improving model training efficiency. Therefore, the authors should analyze the relationship between the number of training iterations and accuracy.
>
> Thank you for bringing up this question. We clarify the training protocol and fairness as follows.
>
> 1. **Standard Practice in Pruning and DCQ**
>    - As stated in lines 445–450 of the paper, both traditional dataset pruning methods and our approach
>      follow the **standard setting where the total number of training iterations remains fixed**.
>    - Therefore, neither prior pruning methods nor ours aim to improve training efficiency.
>
> 2. **Fairness of Comparison**
>    - Accuracy is compared under the **same compression ratio**, making the evaluation fair and aligned
>      with existing pruning literature.
>
> 3. **Performance Under Reduced Training**
>    - Although our main experiments use the standard **40,000-iteration protocol**,
>      Table B1 shows that even with **substantially fewer epochs**
>      (e.g., 50 epochs ≈ 9,750 iterations),
>      our method still **outperforms SOTA pruning methods trained for ~40,000 iterations**.
>    - This demonstrates the **robustness and stability** of our approach.
>
> 4. **Revision Note**
>    - Table B1 and the corresponding explanations have been **revised and updated** in Appendix F.
>
>
>
>
> Table B1: Different number of training epochs on CIFAR-10 on ResNet-18 under 1-bit color quantization.
>
> |  number of epochs | number of Training Iterations| Acc |
> |:-------|:---:|:---:|
> | 50  |9,750| 75.01 |
> | 100  | 19,500 | 76.19 |
> | 125 | 24,375 | 77.69 |
> | 150 | 29,250 |78.57 |
> |200 | 39,000 | 79.90|
>
>
>
> >2. Some recent state-of-the-art or more related methods are missing, such as [1,2,3,4,5].
>
> Thank you for your question. Table B2 shows the comparison of our algorithms with recent SOTA-related methods [1,2,3,5]. Table B3 shows the comparison of our algorithms with  AutoPalette[4]. All these tables have been revised and updated for the paper revision to Appendix F.
>
> **Table B2.** Comparison of different dataset pruning algorithms on CIFAR-10 (ResNet-18) and ImageNet-1k (ResNet-34).
>
> | **CIFAR-10 (ResNet-18)** | 80% | 87.5% | 92% | 96% |  | **ImageNet-1k (ResNet-34)** | 80% | 87.5% | 92% | 96% |
> |:--------------------------|:---:|:---:|:---:|:---:|---|:-----------------------------|:---:|:---:|:---:|:---:|
> | ADQ [5]   | 90.40 | 87.51 | 86.32 | 75.99 |  | ADQ [5]   | 65.31 | 60.91 | 47.02 | 32.19 |
> | DQAL [3]  | 90.20 | 87.61 | 85.30 | 75.80 |  | DQAL [3]  | 62.21 | 60.41 | 46.73 | 31.89 |
> | DQ [2]    | 89.40 | 87.70 | 84.25 | 77.92 |  | DQ [2]    | 64.30 | 60.45 | 47.17 | 32.27 |
> | DUAL [1]  | 91.42 | 88.99 | 86.15 | 74.83 |  | DUAL [1]  | 66.50 | 61.15 | 46.55 | 33.83 |
> | **Ours**  | **94.39** | **91.02** | **89.15** | **79.90** |  | **Ours** | **66.99** | **62.02** | **49.69** | **35.95** |
>
> **Table B3.** Comparison between our algorithm with AutoPalette on CIFAR-10 and CIFAR-100(ConvNetD3).
>
> |            | **CIFAR-10** |         | **CIFAR-100** |        |
> |-----------:|:------------:|:-------:|:-----------------:|:------:|
> |            | IPC=10       | IPC=50  | IPC=10            | IPC=50 |
> | AutoPalette     |  74.3         | 79.4    |  52.6             | 53.3  |
> | **Ours**    | **77.9**         | **82.3**    | **58.9**             | **61.7**  |

---

> > ### Author Response · Authors · 2025-11-24
> >
> > >3. Color quantization, in a sense, functions similarly to data augmentation. What if dataset pruning is also combined with a similar data augmentation strategy? If such a combination also yielded strong performance, the contribution of this paper would be somewhat diminished, because dataset pruning plus data augmentation would not only reduce storage requirements but also decrease the number of training iterations while maintaining performance.
> >
> >  Thank you for highlighting this concern. We clarify the distinction and provide supporting evidence below.
> >
> > 1. **DCQ vs. Data Augmentation**
> >    - DCQ replaces each original image with its quantized, cluster-shared version.
> >    - The model is trained **solely on compressed images**, reducing dataset storage and memory.
> >    - Augmentation adds **extra perturbed samples** but does **not reduce dataset size, bit-depth, or replace originals**.
> >    → Therefore, DCQ and augmentation serve fundamentally different purposes.
> >
> > 2. **Fair Comparison**
> >    - We apply the same augmentation settings (RandomCrop + RandomHorizontalFlip, following [6])
> >      to DCQ and all pruning baselines.
> >    - Thus, the performance gains of DCQ come from the proposed quantization strategy,
> >      not from differences in augmentation.
> >
> > 3. **Complementary Effects**
> >    - As shown in Table B4, adding standard augmentations (ColorJitter, GaussianBlur)
> >      on top of DCQ, **further improves performance**.
> >    - This demonstrates that DCQ is complementary to augmentation rather than a substitute.
> >
> >
> > Table B4. Combining other dataset augmentation algorithms on CIFAR-10 on ResNet-18 under 1-bit quantization.
> > | Method | 80% | 87.5% | 92% | 96% |
> > |:-------|:---:|:---:|:---:|:---:|
> > | Original setting  |94.39|  91.02 |  89.15 | 79.90|
> > | Original setting + ColorJitter  | 95.02|  92.44 |  90.45 | 80.95 |
> > | Original setting + GaussianBlur | 95.47|  92.95 |  91.47 | 81.75 |
> >
> >
> > [6] Zheng, Haizhong, et al. "Coverage-centric coreset selection for high pruning rates." arXiv preprint arXiv:2210.15809 (2022).

---

### Official Review · Reviewer_ivQG · 2025-11-02

**Soundness:** 3
**Presentation:** 3
**Contribution:** 2
**Rating:** 4
**Confidence:** 4

**Summary:**

The paper proposes a novel method, Dataset Color Quantization (DCQ), that reduces dataset storage by compressing color information rather than discarding samples. Unlike traditional pruning or distillation methods that lower dataset size by removing images, DCQ targets color-space redundancy within images to achieve compression while preserving critical training information. The proposed method does not offer significant training or inference time savings. The experiments are based on classification tasks over CIFAR-10/100, TinyImageNet, and ImageNet-1k.

**Strengths:**

1. The paper's method is clearly stated, and the paper is easy to follow.

2. The proposed method is intuitive and proven to be effective under the defined settings.

**Weaknesses:**

1. I am not very clear about the comparison with direct image quantization, especially as the size of the dataset increases. For example, if extending the experiment to ImageNet 22k, will the proposed method still outperform the direct image quantization?

2. As the classification task is highly abstract, the proposed method could work. However, it is not clear if the method still works for dense prediction tasks such as image segmentation and object detection. It would make the paper stronger if there were some experiments on generative tasks. Besides, it is also interesting to see if the proposed method could work together with model quantization, which could lead to significant model training and inference speed-up.

**Questions:**

Please refer to the weakness part.

---

> ### Author Response · Authors · 2025-11-24
>
> >1. I am not very clear about the comparison with direct image quantization, especially as the size of the dataset increases. For example, if extending the experiment to ImageNet 22k, will the proposed method still outperform the direct image quantization?
>
> Thank you for your question. Table A1 reports the computational complexity of DCQ, which scales linearly with the size of the dataset. Since standard image quantization methods such as JPEG are also well-known to have linear complexity (requiring only a single pass over the dataset), both approaches exhibit the same linear scaling behavior. Therefore, DCQ inherits the same level of scalability as direct image quantization and is well-suited for large-scale datasets. This table has been revised and updated for the paper revision to Appendix G.
>
> Table A1: Computational complexity of DCQ on large-scale datasets. BackboneFLOPs denotes the FLOPs required for one forward inference of a 224×224 image.
>
> | Dataset        | #Classes (C) | Images per Class (IPC) | Total Images (N = C × IPC) | Complexity                         |
> |----------------|--------------|-------------------------|------------------------------|------------------------------------|
> | ImageNet-1K    | 1,000        | ≈1200                   | ≈1.2M                       | O(1,000 × 1200 × BackboneFLOPs)    |
> | ImageNet-21K   | 21,000       | ≈700                    | ≈14.7M                      | O(21,000 × 700 × BackboneFLOPs)    |
>
> >2. As the classification task is highly abstract, the proposed method could work. However, it is not clear if the method still works for dense prediction tasks such as image segmentation and object detection. It would make the paper stronger if there were some experiments on generative tasks. Besides, it is also interesting to see if the proposed method could work together with model quantization, which could lead to significant model training and inference speed-up.
>
> Thank you for your valuable feedback. Table A2 and Table A3 show that our algorithm outperforms other dataset pruning methods on both object detection and image segmentation tasks. Table A4 shows that our algorithm outperforms other dataset pruning algorithms on generative tasks.  All these tables have been revised and updated for the paper revision to Appendix F.
>
> Table A2. Different Dataset pruning algorithms on image segmentation on MS COCO, AP₅₀, model trained on the full dataset is 55.20%.
> | Method |  80%  | 87.5% | 92% | 96% |
> |:-------|:---:|:---:|:---:|:---:|
> | Random  |45.60| 42.07 | 38.15 | 25.99|
> | CCS  |43.68| 39.91 | 35.58 | 26.46 |
> | EL2N |38.98 | 36.12 | 29.91 | 24.26 |
> | **Ours**|**47.38** | **44.29** | **41.68** | **28.77** |
>
> Table A3. Different Dataset pruning algorithms on object detection on MS COCO, AP₅₀ᵇᵇ, model trained on the full dataset is 58.30%.
> | Method | 80% | 87.5% | 92% | 96% |
> |:-------|:---:|:---:|:---:|:---:|
> | Random  |48.80| 44.07 | 40.15 | 31.89|
> | CCS  | 46.71 | 42.89 |37.08| 30.15 |
> | EL2N | 40.05 | 39.15 | 31.48 | 27.36 |
> | **Ours**| **50.78** | **46.29** |**42.08** | **33.37** |
>
> Table A4. Comparison of dataset pruning algorithms on the CIFAR-10  PSLD [1] generative task [1]. Results are evaluated by FID (↓), where lower values indicate better performance.
>
> | Method | 80% | 87.5% | 92% | 96% |
> |:-------|:---:|:---:|:---:|:---:|
> | Random  |16.8 | 39.4 | 50.2 | 119.8|
> | CCS  | 15.7 | 37.8  |52.8| 121.1|
> | EL2N | 18.1 | 45.1 | 61.4 | 138.3 |
> | **Ours**| **11.9** | **21.3** |**42.2** | **113.5** |
>
> As discussed in Section 4.3, dataset color quantization mainly reduces dataset storage rather than training time. Since our method does not modify the neural network structure, it is fully compatible with existing model quantization techniques and can be jointly applied to achieve training and inference speed-up without affecting model quantization performance.
>
> [1] Pandey, K., & Mandt, S. (2023). A complete recipe for diffusion generative models. In Proceedings of the IEEE/CVF International Conference on Computer Vision (pp. 4261-4272).

---

### Meta-Review · Area_Chair_ujud · 2025-12-30

**Summary:**

This paper proposes a color quantization method for datasets, solving the dataset-level compression problem through a holistic training framework. This is a fundamental problem in image processing and computer vision, and experimental data demonstrates the effectiveness of the method.

This paper was on the verge of acceptance or rejection, attracting the attention of reviewers during the discussion phase. Some reviewers provided relatively positive feedback. Furthermore, the paper effectively addressed the reviewers' concerns during the review response and summary phases. The AC believes that most of these concerns have indeed been resolved to some extent.

**Reviewer Concerns:**

+ Reviewer ivQG expressed concerns about larger datasets, including multiple tasks. The authors supplemented their findings with experiments on larger datasets, including segmentation, detection, and other tasks, and included these in relevant tables.

+ Reviewer FTHp raised comparisons with recent related work, particularly listing five state-of-the-art (SOTA) methods published between 2023 and 2025, which the AC considers very important. A careful review of the comparisons with these methods revealed that the authors' method achieved certain results on representative datasets such as ImageNet1k and CIFAR10. The AC urged the authors to incorporate these key revisions into the main text. Reviewer KUXK expressed concerns about the authors' originality, experimental evaluation, and time efficiency. The authors addressed most of these concerns, particularly by adding numerous tests to improve experimental time efficiency.

+ Reviewer NGTe gave a relatively positive score, raising a few questions about visualization.

+ Reviewer Jn5n initially expressed significant concerns about experimental fairness and the main experiments. The authors provided detailed and comprehensive answers to these concerns regarding experimental baselines, external data leakage, and spatial guidance.

**Reviewer Scores:**

Summarizing the reviewers' opinions, after the discussion, two reviewers offered relatively positive feedback, while three gave borderline negative feedback. The AC carefully reviewed these comments and indeed believed that most of the key concerns were addressed in the authors' rebuttal. The authors' responses were very detailed, meticulously listing the weaknesses raised by the reviewers and making revisions to the key sections of the text. The area chair acknowledged these efforts and considered the paper borderline for acceptance.

---

### Decision · Program_Chairs · 2026-01-26

Accept (Poster)